# Control of bacterial quorum threshold for metabolic homeostasis and cooperativity

Eunhye Goo,[1,2] Ingyu Hwang[1,2]

**ABSTRACT**    Many *Proteobacteria* employ an acyl-homoserine lactone (AHL)-mediated quorum sensing (QS) system to control diverse social behaviors in a cell density-dependent manner. Various QS modulation mechanisms securing QS initiation at high cell density have been described. However, how QS bacteria determine the quorum threshold is less well known than expected, and little is known about how their physiological and social traits are affected by problems making such early decisions. Here, we show that the RNA-binding protein TofM binds to the mRNA of the QS signal synthase gene *tofI* and prevents QS signal biosynthesis at low cell density (LCD), thereby defining a stringent QS threshold for the rice pathogen *Burkholderia glumae*. The *tofM* mutant produced significant amounts of QS signals at LCD, resulting in a reduced growth rate due to advanced metabolic slowing and metabolic imbalance. When *tofM* mutants were grown in closed batch culture, mutants of *qsmR*, encoding a QS-dependent master regulator, spontaneously emerged. The same type of *qsmR* mutation was observed when low-density wild-type cells were cultured at AHL concentrations above the QS threshold. These data showed that translational control of the QS signal synthase gene at LCD is a stringent mechanism to maintain metabolic homeostasis and cooperativity in *B. glumae*. Our findings reveal that the bacterial genetic system has diversified to ensure the social activity of QS bacteria, as well as the possible consequences of QS bacteria at LCD encountering environments in which signals generated by their natural neighbors exceed their QS threshold.

**IMPORTANCE**    The mechanisms used by various bacteria to determine whether their density is sufficient to meet the QS threshold, how stringently bacterial cells block QS initiation until the QS threshold is reached, and the impacts of low-density bacterial cells encountering conditions that exceed the QS threshold are longstanding gaps in QS research. We demonstrated that translational control of the QS signaling biosynthetic gene creates a stringent QS threshold to maintain metabolic balance at low cell densities. The emergence of non-cooperative cells underlines the critical role of stringent QS modulation in maintaining the integrity of the bacterial QS system, demonstrating that a lack of such control can serve as a selection pressure. The fate of quorum-calling cells exposed to exceeding the QS threshold clarifies QS bacteria evolution in complex ecosystems.

**KEYWORDS**    acyl-homoserine lactone, low cell density, RNA binding protein TofM, cooperativity

Address correspondence to Eunhye Goo, sourire11@snu.ac.kr.

The authors declare no conflict of interest.

See the funding table on p. 15.

Quorum sensing (QS) is known as a cell-cell communication process in which bacteria monitor the quantity of small, diffusible signal molecules in their environment to indirectly assess their population density and coordinate gene expression (1). This process can activate or repress a variety of genes involved in physiological processes such as bioluminescence, exoenzyme production, motility,

biofilm formation, and primary metabolism (2–5). One interesting aspect of QS is the threshold at which it becomes activated, which remains a subject of significant scientific interest and has not yet been fully elucidated for most QS bacteria.

Various factors, including temperature, pH, nutrient availability, and the presence of signal molecules released by QS bacteria in the environment, can influence the QS threshold (1–3, 6). In addition, the presence of receptors or transcriptional regulators that bind to signaling molecules can impact the QS threshold (7–13). It is important to note that the mechanisms governing QS threshold determination may exhibit variability among bacterial species and ecological niches.

One well-described mechanism for QS modulation involves anti-activators that sequester QS signal receptors or transcriptional regulators (7–13). This anti-activation process modulates the QS threshold and delays QS-mediated gene expression (7–13). For example, TraM was the first QS anti-activator to be characterized; it sequesters TraR to delay QS-mediated Ti plasmid conjugal transfer in *Agrobacterium tumefaciens* (7, 8). TraM-mediated anti-activation in *A. tumefaciens* is a good example of specific conditions, in this case conjugal opines, in the environment of QS bacteria that are involved in QS modulation (7, 8). Anti-activation has also been observed in the human pathogen *Pseudomonas aeruginosa* (9–13). In *P. aeruginosa*, the anti-activators QslA and QteE physically interact with LasR and RhlR to attenuate early activation of the QS regulon (9, 10). QscR in *P. aeruginosa* sequesters acyl-homoserine lactone (AHL) signals from LasR *via* the formation of heterodimers of QscR and LasR (11–13). However, QscR also binds AHL signals, resulting in the repression of genes that are activated by LasR or RhlR and thus delaying their expression (11–13). These anti-activators also play an important role in disrupting self-sensing (13). However, how widespread QS modulation is among QS bacteria remains unknown.

While it has been suggested that even a single cell in a small volume can constitute a quorum (14), the QS threshold is generally reached at high cell densities although cells sense AHL concentration, which is correlated with cell density under certain conditions (15). Empirical studies have shown that activation of QS at high cell densities is generally more beneficial to bacteria (16, 17). While this observation applies to some bacterial species, it is important to note that the specific conditions and regulatory mechanisms can vary among different microbes, especially those inhabiting diverse ecological niches (18, 19). For these reasons, most studies on QS-dependent gene regulation have focused on how QS bacteria behave when they are above the QS threshold. Meanwhile, little attention has been paid to metabolism and control of gene expression in low-density cells that have not reached the QS threshold. Furthermore, whether exposure of low-density cells to high-level signals has a negative effect on the coordinated behaviors exhibited by bacteria to ensure the overall benefit of the population remains unclear.

In the case of *Burkholderia glumae*, the causative agent of rice panicle blight, the QS system involves a LuxI-R-type system in which TofI synthesizes the small diffusible molecules *N*-hexanoyl-homoserine lactone (C6-HSL) and *N*-octanoyl-homoserine lactone (C8-HSL) (20). The complex formed from C8-HSL and its cognate receptor TofR activates expression of the transcriptional regulator QsmR, which, in turn, activates genes associated with motility (21), catalase activity (22), oxalate biosynthesis to neutralize ammonia toxicity (23), and metabolic slowing to ensure metabolic homeostasis and survival in crowded conditions (5). The QS in *B. glumae* serves as a mechanism to anticipate the stationary phase or life at the carrying capacity of a population by activating the expression of cytoplasmic enzymes, altering cellular metabolism, and producing a shared resource or public good (in this case, oxalate) (23). The QS onset density for *B. glumae* under laboratory conditions was $OD_{600} = 1.5$, equivalent to $1 \times 10^9$ colony forming unit (CFU)/mL, and the expression of QS-regulated genes was not detected until reaching the QS threshold (23). Notably, this threshold was significantly higher than the $OD_{600}$ range of 0.2–0.4, equivalent to $1 \times 10^8$ CFU/mL, which induces QS-dependent bioluminescence in *Vibrio fischeri* (24). It was also higher than the $OD_{600}$ of 0.2, at which expression levels of the QS-dependent *lasB* gene can be measured by

GFP fluorescence in *P. aeruginosa* (13). We were interested in why *B. glumae* requires higher cell density than other QS bacteria to meet the QS threshold. Our aim was to determine the molecular mechanisms underlying the abnormally high level of C8-HSL production in the *tofM* mutant, as previously reported (25). In this study, we identified the mode of action involved in setting the TofM-mediated QS threshold and found that both *tofM* mutation and early exposure of low-density wild-type cells to signal levels above the QS threshold caused slow growth, metabolic disorder, and the emergence of non-cooperative cells.

Our findings illustrate how QS systems have evolved to determine the QS threshold and demonstrate the importance of stringent QS modulation to the cooperation of bacterial cells at LCD. Understanding the mechanisms that control QS threshold determination and their impacts on bacterial behavior and fitness elucidates the complex dynamics of QS-mediated interactions in bacterial communities.

## RESULTS

### Stringent control of TofM-mediated QS initiation

To determine the QS threshold of *B. glumae* BGR1, we evaluated the levels of C8-HSL and C6-HSL produced at approximately $2 \times 10^7$, $3 \times 10^8$, and $2 \times 10^9$ CFU/mL *via* thin layer chromatography assays using *Chromobacterium violaceum* CV026 as an indicator. Wild-type BGR1 at a density of $2.47 \times 10^9$ CFU/mL produced detectable by eye amounts of C8-HSL and C6-HSL in our assays, but no visible positive spots were observed at $2.37 \times 10^7$ CFU/mL and $3.23 \times 10^8$ CFU/mL (Fig. 1A). By contrast, the *tofM* null mutant generated *via* EZ-Tn*5* insertion in the coding region as shown in Fig. 1B produced C8-HSL and C6-HSL at eye-detectable levels at a density of $2.83 \times 10^7$ CFU/mL (Fig. 1B). The *tofM* mutant produced substantially more C8-HSL and C6-HSL at $3.73 \times 10^8$ CFU/mL and $2.6 \times 10^9$ CFU/mL than the wild type when cell growth and preparation of AHL were identical for each strain (Fig. 1A). AHL production of the *tofM* mutant was restored to wild-type levels when pTOFM4 carrying *tofM* and its promoter region was provided *in trans* (Fig. 1A). This indicated that TofM exerts a negative control on the average AHL production in *B. glumae*.

### TofM is an RNA-binding protein targeting mRNA of *tofI*

To determine the mechanisms involved in TofM-mediated regulation of AHL production, the expression levels of *tofI* and *tofR* were measured in the wild type and the *tofM* mutant *via* quantitative reverse transcription polymerase chain reaction (qRT-PCR). Transcription levels of *tofI* and *tofR* did not significantly differ between the wild type and the *tofM* mutant at either $3 \times 10^8$ CFU/mL or $2.4 \times 10^9$ CFU/mL (see Fig. S1 in the supplemental material). Next, we measured TofI and TofR levels in the wild type and the *tofM* mutant through immunoblot analysis with anti-TofI and anti-TofR antibodies. The level of TofI was higher in the *tofM* mutant than in the wild type at both $3 \times 10^8$ CFU/mL and $2.4 \times 10^9$ CFU/mL (Fig. 1C). The intense lower band observed in Fig. 1C left panel was non-specific. TofI levels in the *tofM* mutant were restored to wild-type levels upon complementation with pTOFM4 (see Fig. S2 in the supplemental material). By contrast, levels of TofR did not significantly differ between the wild type and the *tofM* mutant (Fig. 1D). These results indicate that TofM post-transcriptionally regulates the expression of the *tofI* gene.

To investigate the mechanisms of post-transcriptional regulation of *tofI* by TofM, we predicted the secondary structure of the promoter region of *tofI*. We observed a hairpin structure covering the putative ribosome binding site in the 5′ untranslated region (UTR) of *tofI* (Fig. 2A). To further test whether TofM binds to the mRNA of *tofI*, we performed an electrophoretic mobility shift assay using TofM and biotinylated *tofI* mRNA synthesized *in vitro*. TofM bound to the 763-nucleotide (nt) long *tofI* mRNA, which includes the 5′UTR and the entire coding region of *tofI* (Fig. 2B). The binding of labeled *tofI* mRNA was hindered after the 100-fold addition of unlabeled 763-nt *tofI* mRNA (Fig. 2C). TofM also

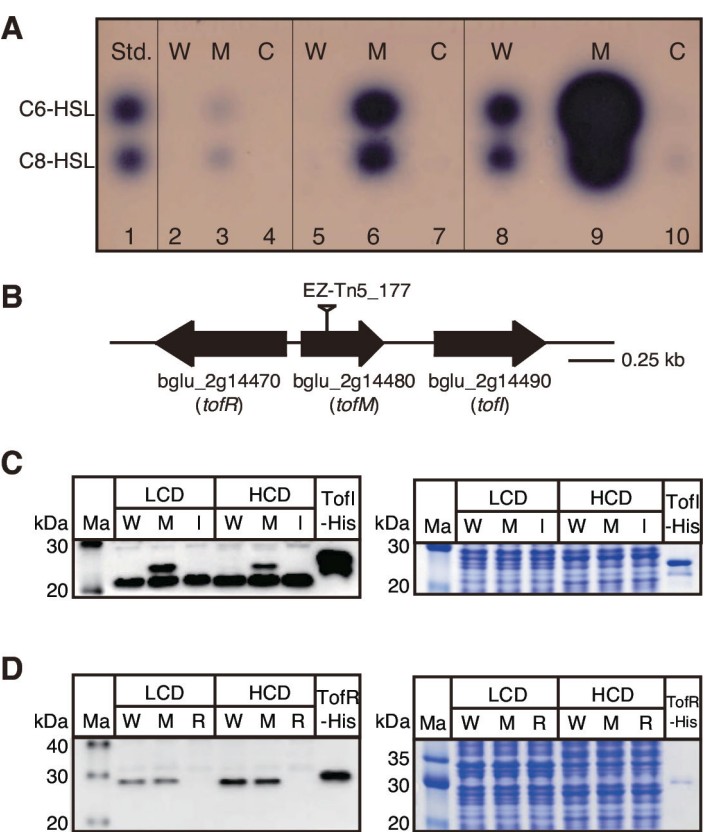

**FIG 1** Autoinducer production and TofI expression levels in the *tofM* mutant. (A) Prepared autoinducers were separated on thin-layer chromatography plates overlaid with the *Chromobacterium violaceum* CV026. 1; synthetic standards of 25 pmol C6-HSL and 1 nmol C8-HSL, 2; wild-type BGR1 at $2.37 \times 10^7$ CFU/mL, 3; *tofM* mutant at $2.83 \times 10^7$ CFU/mL, 4; complementation strain of the *tofM* mutant at $1.87 \times 10^7$ CFU/mL, 5; wild-type BGR1 at $3.23 \times 10^8$ CFU/mL, 6; *tofM* mutant at $3.73 \times 10^8$ CFU/mL, 7; complementation strain of the *tofM* mutant at $3.03 \times 10^8$ CFU/mL, 8; wild-type BGR1 at $2.47 \times 10^9$ CFU/mL, 9; *tofM* mutant at $2.6 \times 10^9$ CFU/mL, 10; complementation strain of the *tofM* mutant at $1.93 \times 10^9$ CFU/mL. (B) Gene map of the QS system in *B. glumae* BGR1, shown with gene names below the open reading frame (ORF). The vertical bar above the ORF indicates the position of EZ-Tn5 insertion. (C) The amount of TofI in each strain was determined *via* Western blotting analysis using anti-TofI antibody. (D) Western blotting using anti-TofR antibody indicated similar levels of TofR in both the wild type and *tofM* mutant. Ma; molecular marker, LCD; low cell density ($3.0 \times 10^8$ CFU/mL), HCD; high cell density ($2.4 \times 10^9$ CFU/mL), W; wild type, M; *tofM* mutant, C; *tofM⁻/tofM*, I; *tofI* mutant, R; *tofR* mutant. Sodium dodecyl sulfate-polyacrylamide gel electrophoresis image, stained with Coomassie Brilliant Blue R-250 (Sigma) and showing that samples were loaded equally in all lanes. All data are representative of triplicate experiments.

bound to 306-nt length *tofI* mRNA that contained the 5′UTR and some part of coding region of *tofI* near the 5′UTR, and the binding of labeled *tofI* mRNA and TofM was hampered by the 100-fold addition of unlabeled 306-nt *tofI* mRNA (Fig. 2D and E). TofM did not bind to the nonspecific targets of labeled 809-nt and 300-nt length of *katE* mRNA (Fig. 2C and E; see also Fig. S3 in the supplemental material).

## Reduced growth rate of the *tofM* mutant attributed to an advanced metabolic slowing

The *tofM* mutant and the wild type supplemented with 10 µM C8-HSL had significantly lower growth rates than the wild type grown in lysogeny broth (LB) medium (Fig. 3A). To determine whether the slow growth of the *tofM* mutant and wild type grown in the

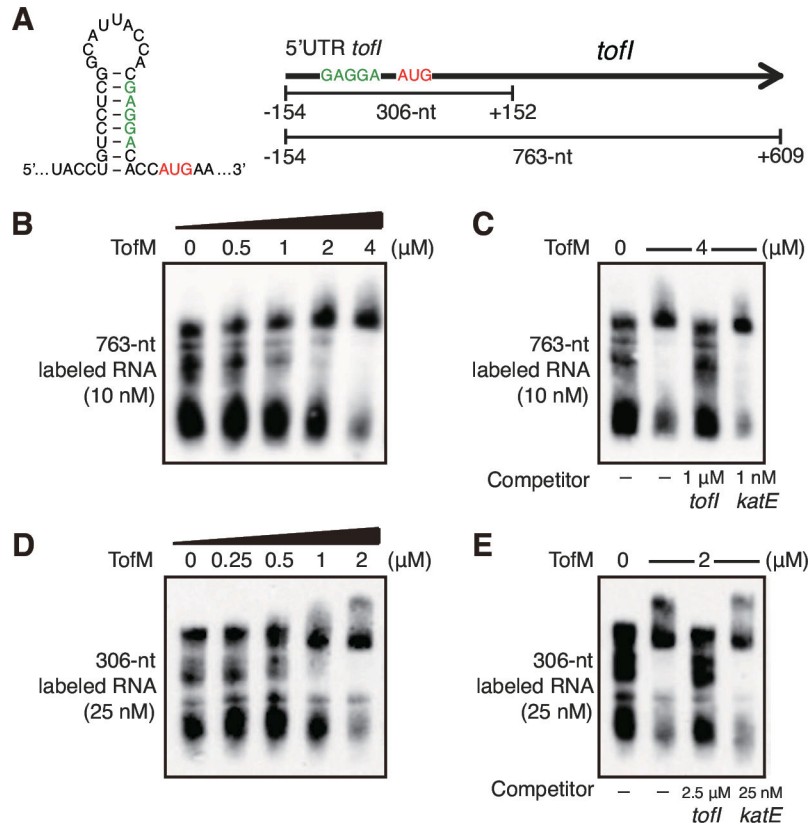

**FIG 2** Binding of TofM to *tofI* mRNA. (A) Map of the *tofI* ORF showing the predicted hairpin structure and the putative TofM-binding sites, including the ribosome binding site (green). The secondary structure of the promoter region of *tofI* was predicted using MXfold software at the web server (http://www.dna.bio.keio.ac.jp/mxfold/). This software is based on the source code of CONTRAfold and is released under the MIT license. The transcription start codon AUG is highlighted in red. (B) Binding assays were conducted using an *in vitro*-transcribed 763-nt RNA corresponding to the 5′UTR and an internal region of the *tofI* ORF (nucleotides −154 to +609 relative to the start codon). Biotin-labeled RNA (10 nM) was incubated with increasing concentrations of TofM. (C) The binding of 4 µM TofM to labeled *tofI* mRNA (10 nM) was blocked in the presence of the specific competitor, unlabeled *tofI* mRNA (1 µM). The nonspecific competitor of unlabeled *katE* mRNA (10 nM) was used. (D) Binding assays were performed using an *in vitro*-transcribed 306-nt RNA for the binding assay (nucleotides −154 to +152 relative to the start codon). Biotin-labeled RNA (25 nM) was incubated with increasing concentrations of TofM. (E) The binding of 4 µM TofM to labeled *tofI* mRNA (25 nM) was blocked in the presence of the specific competitor, unlabeled *tofI* mRNA (2.5 µM). The nonspecific competitor of unlabeled *katE* mRNA (25 nM) was used. The RNA binding assays were repeated three times with three independent replicates and were repeatable.

presence of high concentrations of C8-HSL was due to metabolic slowing in the early growth stage, the levels of amino acids remaining in the cultures of each strain were measured. This assessment is particularly relevant since QS acts as a metabolic brake, ensuring metabolic homeostasis in the primary metabolism of *B. glumae* (5). Of the 19 amino acids tested, 17 amino acids, all excluding lysine and histidine, were consumed more slowly by the *tofM* mutant and wild type supplemented with 10 µM C8-HSL than by the wild type without supplementation (Fig. 3B through D; see also Fig. S4 in the supplemental material). The presence of pTOFM4 allowed the growth and consumption of amino acids by the *tofM* mutant to recover to the rates of the wild type (Fig. 3; see also Fig. S4 in the supplemental material). These results indicate that the slow growth rates of the *tofM* mutant and wild type supplemented with 10 µM C8-HSL in LB were due to advanced metabolic slowing.

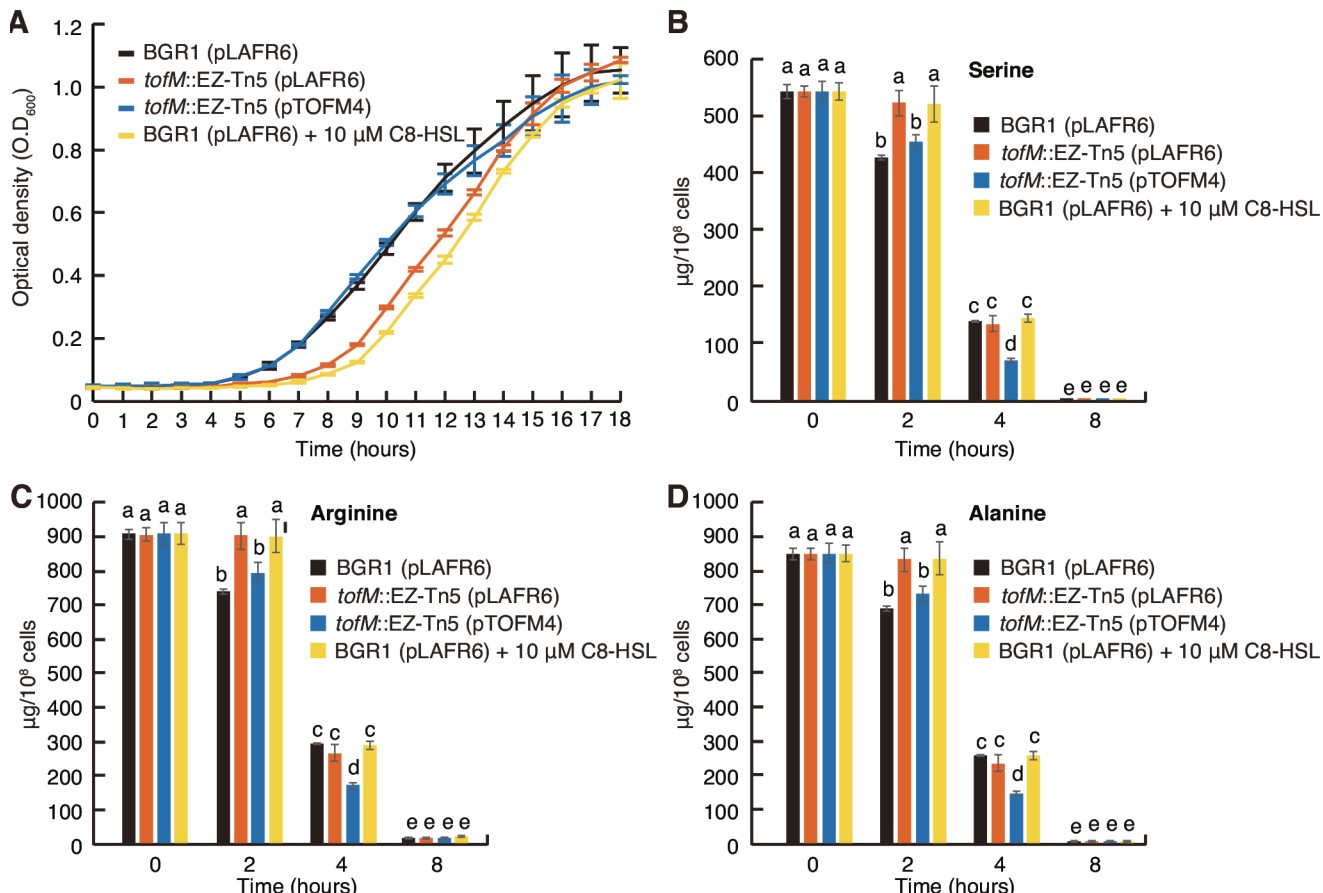

**FIG 3** Decreased growth rate and metabolic slowing in the *tofM* mutant at the early growth stage. (A) Optical density at 600 nm (OD$_{600}$) of the wild type, *tofM* mutant, complementation strain of the *tofM* mutant, and the wild type supplemented with 10 µM C8-HSL was measured at designated time points after subculturing of the seed culture through 20-fold dilution from OD$_{600}$ of 0.05. The cultures were grown in a microplate containing LB broth at 37°C with orbital shaking at 250 rpm and OD$_{600}$ was measured using a microplate reader (Omega, BMG LabTech). (B) The levels of serine, (C) arginine, and (D) alanine in the culture supernatants of the wild type, *tofM* mutant, complementation strain of the *tofM* mutant, and the wild type supplemented with 10 µM C8-HSL were determined by high-performance liquid chromatography (HPLC, Thermo Dionex) at designated time points. Amino acid levels were normalized through division by bacterial density (CFU/mL) and expressed as µg/10$^8$ cells. The data show the mean ± standard deviation of triplicate experiments. The letters (a, b, c, d, and e) above each mean represent significant differences based on a one-way ANOVA, followed by Tukey's post-hoc analysis. A separate ANOVA was conducted for each time point in the figures, rather than a single ANOVA covering all measurements at all time points. A value of $P < 0.05$ represented significant differences among strains.

## Emergence of non-cooperative *qsmR* mutants from the *tofM* mutant

To understand why *B. glumae* stringently controls QS initiation, we tested whether disruption of the quorum threshold in the *tofM* mutant or low-density *B. glumae* cells supplemented with 10 µM C8-HSL has any adverse effects on the collective behaviors exhibited by bacteria for the overall benefit of the population. During growth in closed-batch culture, morphologically distinct spontaneous mutants arose from the *tofM* mutant 5 days after subculturing, and the proportion of the mutant increased to 50% of the total population (Fig. 4A through C). In the wild-type culture, spontaneous mutants were observed 10 days after subculturing, and the percentage of such mutants in the total population remained less than 1% (Fig. 4B and C). The colony morphology closely resembled that of previously reported QS mutants (26) as depicted in Fig. 4A. In addition, the spontaneous mutants exhibited sensitivity to alkaline pH (see also Fig. S5B in the supplemental material), leading us to investigate the presence of any mutations in QS-dependent oxalate biosynthesis through direct regulation of QsmR (23). We conducted experiments to identify potential mutations in *tofI*, *tofR*, *qsmR*, *obcA*, and *obcB* in the spontaneous mutants. Among five genes that were fully sequenced,

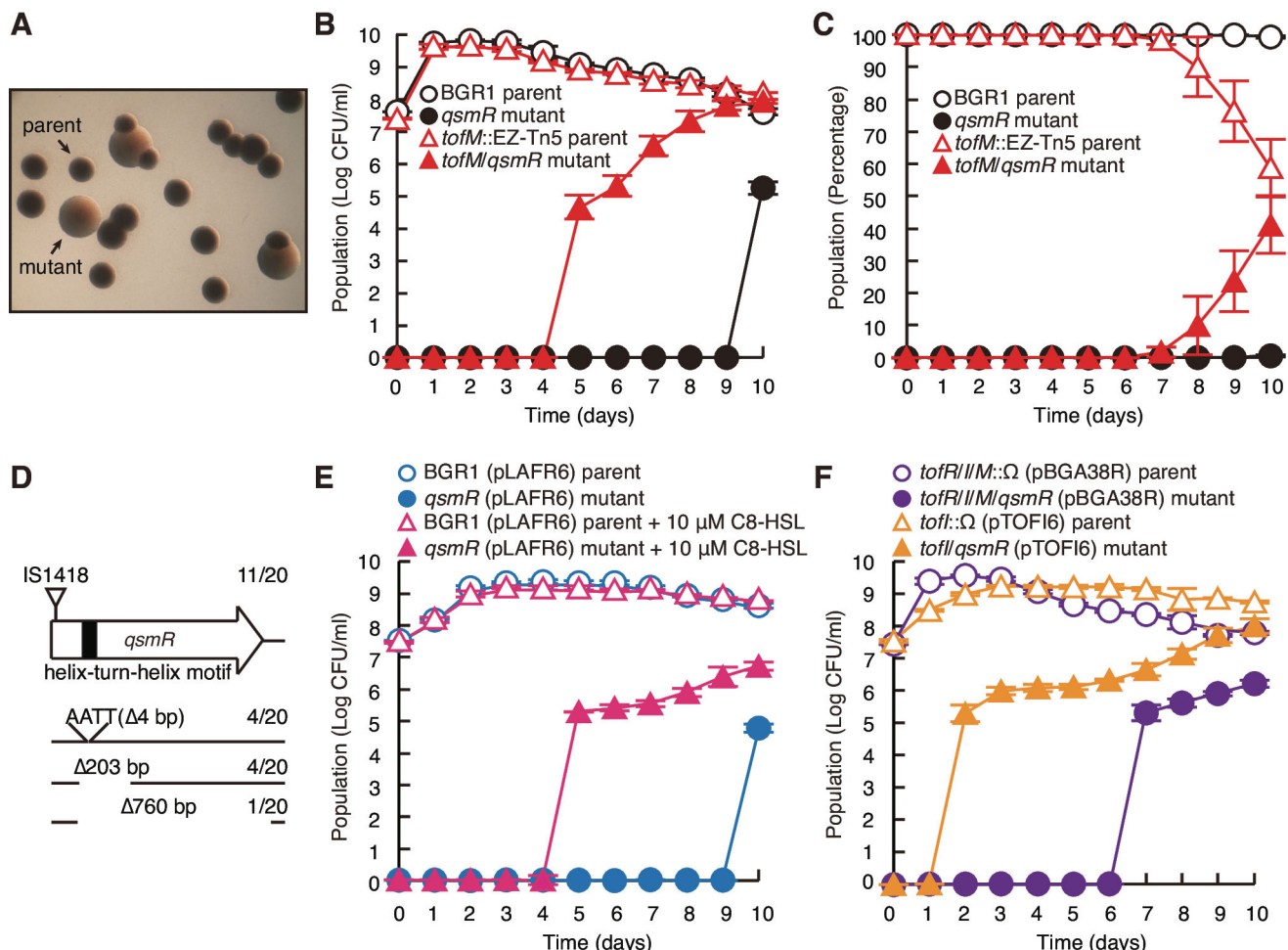

**FIG 4** Appearance of spontaneous mutants for the quorum sensing master regulator (*qsmR*) in a closed batch culture of *B. glumae*. (A) Colony morphologies of the wild-type parent and spontaneous *qsmR* mutant. Morphologically different colonies were photographed using a camera (Canon Powershot G12) connected to a microscope (Nikon SMZ800). (B) The growth of the BGR1 parent (black circle) and *tofM*::EZ-Tn5 parent (red triangle outline) was monitored in LB broth over 10 days of closed batch culture. In addition, the growth of the *qsmR* mutant (black dot) and *tofM/qsmR* mutant (solid red triangle), which derived from the BGR1 parent and *tofM*::EZ-Tn5 parent, respectively, was also monitored. (C) The percentage of each strain in the total population was determined based on growth in LB broth of a 10-day closed batch culture. (D) The positions and types of mutations in the *qsmR* gene were confirmed in the genome of the *tofM/qsmR* mutant. (E) The growth of BGR1 (pLAFR6) parent (blue circle), *qsmR* (pLAFR6) mutant (blue dot), BGR1 (pLAFR6) parent supplemented with 10 µM C8-HSL (pink triangle outline), and *qsmR* (pLAFR6) mutant supplemented with 10 µM C8-HSL (solid pink triangle) in LB broth was compared over 10 days of closed batch culturing. (F) The growth of *tofR/I/M*::Ω (pBGA38R; *tofR/I/M* clone) parent (purple circle), *tofR/I/M/qsmR* (pBGA38R) mutant (purple dot), *tofI*::Ω (pTOFI6; *tofI* clone) parent (orange triangle outline), and *tofI/qsmR* (pTOFI6) mutant (solid orange triangle) in LB broth was compared over 10 days of closed batch culturing. Population density was measured as CFU per mL, and the data are the means of three biological replicates ± standard deviation.

mutations were found only in *qsmR*, including four types of mutations such as IS element (IS1418) insertion and 4 bp, 203 bp, and 760 bp deletions (Fig. 4D). All four types of spontaneous *qsmR* mutants were complemented through insertion of a single copy of *qsmR* along with its promoter region into the chromosome (see Fig. S5 in the supplemental material). When 10 µM C8-HSL was supplied to the wild type, spontaneous *qsmR* mutants similar to those of the *tofM* mutant were observed from 5 days after subculturing and their population size increased over time (Fig. 4E). Spontaneous mutants also appeared from the *tofI* mutant carrying a multi-copy number *tofI* plasmid, illustrating the effect of increased TofI levels, at 2 days after subculturing and reached 48% of the total population (Fig. 4F). In the *tofR/I/M* mutant, which carries a multi-copy-number plasmid of the *tofR/I/M* genes, serving as a control against the *tofI* plasmid, the appearance of spontaneous *qsmR* mutants was observed from 7 days after subculturing, which was

faster than the appearance of spontaneous *qsmR* mutants in *B. glumae* BGR1 carrying an empty vector at 10 days after subculturing (Fig. 4E and F).

## Metabolic imbalance in the *tofM* mutant

To determine the underlying factors driving the mutation of *qsmR* upon early exposure to high levels of QS signals, we investigated oxalate biosynthesis and catalase activity, which are regulated by QsmR. These factors are essential for the survival and adaptation of *B. glumae,* where biosynthesized oxalate neutralizes the alkaline toxicity resulting from deamination in amino acid-based media after the stationary phase (23), and QsmR directly activates the *katG* gene, which responds to oxidative stress in *B. glumae* (22). The *tofM* mutant and the wild type supplemented with 10 µM C8-HSL exhibited higher levels of oxalate biosynthesis than the wild type during 5 days of growth in LB broth (see Fig. S6A in the supplemental material). However, no significant environmental pH differences were observed between the wild type and the *tofM* mutant or the wild type supplemented with 10 µM C8-HSL (see Fig. S6B in the supplemental material). Furthermore, catalase activity was significantly higher in the *tofM* mutant and the wild type supplemented with 10 µM C8-HSL than the wild type during growth in LB broth for 5 days (Fig. 5A). Metabolic activity, which was assessed by measuring the absorbance at 540 nm ($A_{540}$)

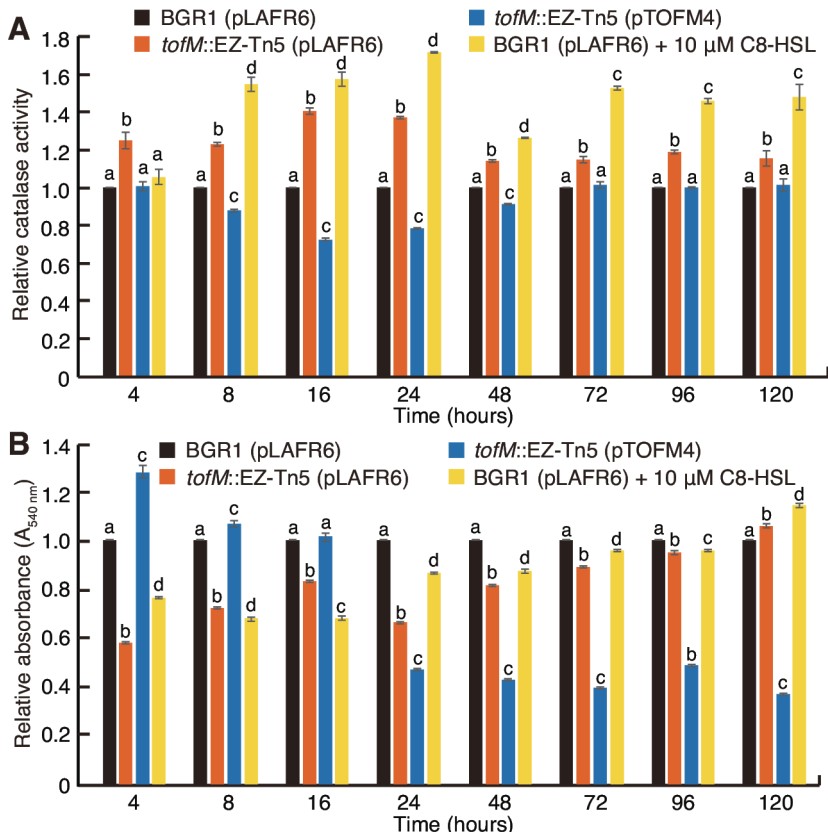

**FIG 5** Catalase and metabolic activities of the *B. glumae* strains. (A) Catalase activity and (B) metabolic activity were measured at designated time points for the wild type, *tofM* mutant, complementation strain of the *tofM* mutant, and the wild type supplemented with 10 µM C8-HSL. The relative catalase activity and relative metabolic activity were calculated by dividing the levels measured in each strain by that of the wild type. The data represent the means of three biological replicates ± standard deviation. The letters (a, b, c, and d) above each mean represent significant differences based on a one-way ANOVA, followed by Tukey's post-hoc analysis. A separate ANOVA was conducted for each time point in the figures, rather than a single ANOVA covering all measurements at all time points. A value of $P < 0.05$ represented significant differences among strains.

of solvent-solubilized formazan, was 40% lower for the *tofM* mutant than the wild type after 4 h of subculturing and was 26% lower after 8 h of subculturing (Fig. 5B). Similarly, the wild type supplemented with 10 µM C8-HSL exhibited 22% to 30% lower metabolic activity during early growth compared to the wild type (Fig. 5B). These differences in metabolic activity gradually decreased over time during culture growth (Fig. 5B). Catalase activity was significantly lower in the spontaneous *qsmR* mutants than in the *tofM* mutant, while the catalase activity of complementation strains for spontaneous *qsmR* mutants was similar to that of the wild type (see Fig. S5C in the supplemental material). The spontaneous *qsmR* mutants had significantly higher metabolic activity than the *tofM* mutant, but this difference gradually decreased as the cultures grew (see Fig. S5D in the supplemental material). Metabolic activity in the spontaneous *qsmR* mutants was complemented with a single copy of *qsmR* and its promoter region (see Fig. S5D in the supplemental material).

## DISCUSSION

In the recent literature, there has been a growing emphasis on understanding the importance of QS-dependent phenotypic heterogeneity in natural settings, highlighting concepts such as the division of labor, specialization, and "bet-hedging" (27). However, it is important to consider the context in which we conducted our experiments, using complex media such as LB and vigorous shaking at 250 rpm. While the term "quorum threshold" is often associated with cell density, which naturally tends to be correlated with AHL concentration (19), our investigations revealed that cells at low density, when exposed to high levels of externally supplied QS signals, experience metabolic imbalance and the emergence of non-cooperative cells during growth. This highlights the critical role of negative regulation of AHL by TofM and the significance of this mechanism. In this context, we refer to the quorum threshold as the cell density, which is correlated with AHL concentration.

It is crucial to understand that QS in *B. glumae* is not optional but essential for survival under specific laboratory conditions, such as growth in glucose-limited LB medium (23). In this environment, QS controls oxalate biosynthesis, a vital process for neutralizing the alkaline pH induced by deamination (23). Unlike the metabolic burden observed in the case of *P. aeruginosa* (28), the effects of the *tofM* mutant were not due to the overproduction of public goods but rather a result of the regulatory role of QS in metabolic homeostasis. The spontaneous *qsmR* mutant indeed functions as a "cheater" in a broad sense, although it does not emerge under conditions necessitating the production of costly public goods (26, 29). Our investigation of the emergence of the *qsmR* mutant from the *tofM* mutant or WT with the addition of AHL was performed to uncover the mechanisms underlying the TofM-mediated QS-negative regulation system in *B. glumae*.

In addition to critical questions of why and how QS bacteria control their thresholds, we also investigated the impacts on genetic and physiological processes when QS bacteria lose control of the threshold and the effects on physiological processes when the QS threshold is influenced by cross-talk with other organisms in the environment. In most previous studies, changes in gene expression and metabolism by QS bacteria have been investigated at high cell densities. In addition, the mechanism through which QS bacteria determine QS threshold has only recently been studied through investigation of QS modulation in *P. aeruginosa* (9–13), although QS modulation was first reported in *A. tumefaciens* (7, 8). In addition to anti-activators as QS modulators, repressors of QS signal synthase gene expression function as QS modulators (30, 31). RsaL represses QS signal production by directly binding to the bidirectional promoter region of *rsaL-lasI* after the stationary phase in *P. aeruginosa*, thereby reducing the expression of *rsaL* and *lasI* through negative feedback mechanisms (30). Similar to RsaL, RsaM functioned as a repressor to inhibit the transcription of two QS signal synthase genes, *pfsI* and *pfvI*, in *Pseudomonas fuscovaginae* (31) and shared 30.4% identity with TofM of *B. glumae* (see Table S1 in the supplemental material). However, RsaL had no significant homology with

RsaM and TofM. Despite some level of identity between TofM and RsaM, TofM did not function as a repressor of the expression of *tofI* and *tofR*.

The global post-transcriptional regulator RsmA modulates AHL production in *P. aeruginosa*, possibly through binding to *rhlI* mRNA to inhibit *rhlI* translation (32). RsmA of *P. aeruginosa* shared 32% identity with TofM of *B. glumae* (see Table S1 in the supplemental material). However, RsmA of *B. glumae* showed no identity with TofM, whether RsmA is involved in QS modulation in *B. glumae* remains unknown. RsmA of *P. aeruginosa* is regulated by GacA/S and sRNA (33, 34), whereas TofM was not associated with any two-component systems. Thus, TofM and RsmA of *P. aeruginosa* do not appear to belong to the same protein group. However, it is conceivable that TofM and RsmA share some homology, as RsmA may function as a translational inhibitor. Although the possible homology between TofM and RsaM in *P. fuscovaginae* is unknown, these three regulators may have evolved divergently in different QS bacteria from a common origin.

Our findings showed that TofM functions by binding to the *tofI* mRNA to prevent the translation of QS signal synthase TofI at low cell density. When the cell density reaches a critical threshold and C8-HSL accumulates, the complex of C8-HSL and TofR activates the transcription of the *tofI* gene through an autoinduction mechanism (20). Consequently, the copy number of *tofI* mRNA becomes higher than that of the TofM, leading to the activation of QS *via* a positive feedback loop in *B. glumae*. Among various QS modulation mechanisms, translational control of *tofI* by TofM in *B. glumae* allowed for more stringent control of QS initiation than other mechanisms, as at least $1 \times 10^9$ CFU/mL was needed to fully initiate QS. This finding raises the question of why *B. glumae* has a more stringent QS threshold compared to other QS bacteria. We believe that targeting translation of the signal synthase gene *tofI* by TofM is a more effective way for delaying target gene induction in *B. glumae* during the early growth stage than interaction with TofR. This process would allow *B. glumae* cells to efficiently produce QS-dependent virulence factors *in vivo* when necessary. Notably, the delay in the expression of virulence factors until later growth stages might significantly influence the pathogenicity dynamics of *B. glumae*. To better comprehend this aspect, future investigations should explore the relevant processes in a more ecologically natural environment. No one has yet investigated how the key virulence factors of *B. glumae* are produced in a QS-dependent manner *in vivo*. Therefore, determining the minimum cell density for virulence factor production *in vivo* would be an interesting research topic. When total populations of *B. glumae* were estimated in a panicle after inoculation, we used CFU/spikelet as the unit. The term panicle in our study refers to the terminal component of the rice tiller and is specific to rice (*Oryza sativa*). A spikelet is the basic unit of the panicle in rice and is commonly used in plant biology terminology. Because the units of CFU/spikelet and CFU/mL are not convertible, QS thresholds should be established *in vivo* and compared to those determined *in vitro*. Such a comparison would clarify how QS and QS-dependent factors studied *in vitro* actually function *in vivo*.

As the determination of the QS threshold is obviously important, we assumed that it must occur in all QS bacteria, but has been confirmed in only a few species. Two explanations are possible for this knowledge gap. First, there may be a completely different mode of action for QS modulation that remains unknown. Second, mechanisms similar to the current models of QS modulation may be widespread among QS bacteria but have not yet been extensively studied or reported. Given the complexity and diversity of bacterial communication systems, it would be unsurprising to see more studies supporting either of these possibilities.

Although the determination of the QS threshold is important for the coordinated functioning of QS bacteria as a social group, the consequences of failing to establish this threshold on the genetic and physiological activities of individual cells remain largely unknown. For *B. glumae*, we believed that QS modulation serves a broader role beyond solely determining the QS threshold. Our current findings demonstrated that QS modulation at LCD plays a vital role in shaping the physiological properties and genetic stability of individual cells. Two lines of evidence support this notion. First, a

lack of QS modulation caused advanced metabolic slowing, which led to delayed growth of the *tofM* mutant due to the slow consumption of amino acids from the culture medium. Second, the *tofM* mutant exhibited higher catalase activity and an altered redox state compared to the wild type. Such metabolic stress appeared to have triggered the emergence of *qsmR* mutants in both the *tofM* mutant and wild type grown with excess amounts of C8-HSL, in accordance with our previous observation that metabolic stress caused mutations in *qsmR* (26, 29). Thus, failure of QS modulation affected the genetic stability of *qsmR*, which is a key regulator for QS-dependent gene expression in *B. glumae*. These findings suggest that QS modulation not only influences the group behavior of QS bacteria but also profoundly affects the genetic dynamics and physiological states of individual cells, highlighting the multifaceted importance of QS modulation in bacterial systems.

When wild-type *B. glumae* was exposed to an excess amount of C8-HSL at LCD, as well as in the *tofM* mutant, QS-dependent genes such as *obcAB*, *flhD*, and *qsmR* were not activated (see Fig. S7 in the supplemental material). This phenomenon has also been observed in *P. aeruginosa*, wherein a majority of QS-dependent genes were not expressed at the early growth stage in response to adding exogenous acyl-HSL signals (35). Interestingly, conditioning the growth medium with non-AHL-producing *P. aeruginosa* enabled the expression of these genes at LCD upon the addition of the AHL signal (36). In *Vibrio fischeri*, it has been noted that the mere presence of a sufficient concentration of cognate AHL, as demonstrated in LuxI-LuxR-like circuits, does not act as a direct trigger for the expression of QS-dependent genes (37). Although the mechanism involved in this process is unclear, assessing whether similar results can be obtained in *B. glumae* would be interesting.

Our findings demonstrate that QS modulation is critical to determine the QS threshold, as well as to maintain metabolic homeostasis and genetic stability at LCD. Exposure of wild-type *B. glumae* cells to excess amounts of C8-HSL caused metabolic and genetic issues similar to those observed in the *tofM* mutant, suggesting that cross-talk is a critical event affecting QS-dependent social activities of QS bacteria in nature. In addition, the emergence of spontaneous *qsmR* mutations from *tofM* mutant due to metabolic stress showed that QS modulation, cellular metabolism, and maintenance of social behavior are closely linked. These discoveries provide valuable insights into the intricate mechanisms underlying the fine-tuned coordination of bacterial behavior in response to variations in cell density and environmental conditions.

## MATERIALS AND METHODS

### Bacterial strains and growth conditions

The bacterial strains and plasmids used in this study are listed in Table S2 in the supplemental material. *B. glumae* and *E. coli* strains were grown at 37°C on Luria-Bertani (LB) agar or in LB broth [0.1% tryptone, 0.5% yeast extract, and 0.5% NaCl (all wt/vol); Thermo Fisher Scientific) with shaking at 250 rpm. The $OD_{600}$ was monitored using a microplate reader while bacterial cells were growing in 200 µL LB broth in each well of a microplate. However, for all other assays, including the determination of CFU, *B. glumae* cells were cultured in 2 mL LB broth in 16 mm × 125 mm, 16.5 mL glass test tubes (PYREX). The following antibiotics were added to the culture medium when needed: rifampicin, 100 µg/mL; trimethoprim, 75 µg/mL; tetracycline, 10 µg/mL; ampicillin, 50 µg/mL; kanamycin, 50 µg/mL; spectinomycin, 100 µg/mL; and chloramphenicol, 34 µg/mL. Synthetic C8-HSL (Sigma) was dissolved in dimethyl sulfoxide and added to the culture medium with a final concentration of 10 µM when needed.

### Generation of the *tofM*::EZ-Tn5 and complementation

The pBGA18 plasmid carrying the *tofM* gene was mutagenized using the EZ-Tn*5* < DHFR-1 > insertion kit (Epicentre). Subsequently, it was mobilized from DH5α into *B.*

*glumae via* conjugation, with pRK2013 serving as a helper plasmid, following the marker exchange method as previously described (20, 21). The EZ-Tn*5* insertion sites were determined through DNA sequencing according to the kit manufacturer's protocols. The *tofM* mutant was confirmed through Southern blot analysis. For genetic complementation of the *tofM* mutant, a 558 bp DNA fragment containing *tofM* and its promoter was generated *via* PCR using the primers EcoRI_ptofM-F and BamHI_stop_His_tofM-R, cloned into pBluescript II SK (+), and named "pTOFM3." A 558 bp DNA fragment obtained from digesting pTOFM3 with *Eco*RI and *Bam*HI was ligated into pLAFR6 to produce pTOFM4. pTOFM4 was mobilized from DH5α into the *B. glumae tofM* mutant *via* conjugation, using pRK2013 as a helper plasmid. The complementation strain of the *tofM* mutant was assessed *via* plasmid extraction.

## Autoinducer assay

To monitor the production of autoinducer from *B. glumae* strains, cells were subcultured by diluting the seed culture 20-fold after adjusting it to an $OD_{600}$ of 0.05. The cultures were sampled at 6, 8, and 10 h after subculturing, corresponding to cell counts of $2 \times 10^7$ CFU/mL, $3 \times 10^8$ CFU/mL, and $2 \times 10^9$ CFU/mL, respectively. Autoinducer extraction and development were performed as described previously (38).

## RNA isolation and quantitative reverse transcription-polymerase chain reaction

Total RNA from wild-type *B. glumae* and the *tofM* mutant, which were grown in LB medium at 37°C for 8 (corresponding to LCD) or 10 hours (corresponding to HCD) after subculturing *via* dilution of the seed culture 20-fold after adjustment to $OD_{600}$ of 0.05, was extracted using the RNeasy mini kits (Qiagen) following the manufacturer's protocol. The total RNA was treated with RNase-free DNaseI (Ambion) to remove genomic DNA. We reverse-transcribed 1 µg quantities of total RNA into complementary DNA using M-MLV Reverse Transcriptase (Promega) for 1 hour at 42°C. The primer pairs used in qRT-PCR are listed in Table S3 in the supplemental material. 16S rRNA served as a positive control. Transcriptional levels were determined using SsoFast EvaGreen Supermix (Bio-Rad) and the CFX96 Real-Time PCR System (Bio-Rad). The thermal cycling parameters were 95°C for 30 s, followed by 40 cycles of 95°C for 5 s and 60°C for 5 s. Each PCR was performed three times, and all data were normalized to the expression level of the 16S rRNA gene using Bio-Rad CFX Manager software.

## Western blot

*B. glumae* cells were harvested from LB medium after 8 (corresponding to LCD) or 10 hours (corresponding to HCD) of growth, following subculture by diluting the seed culture 20-fold after adjusting it to an $OD_{600}$ of 0.05. Total cell lysates were separated by sodium dodecyl sulfate-polyacrylamide gel electrophoresis (SDS-PAGE) and transferred to nitrocellulose membranes. Antibodies against TofR and TofI were generated by immunizing rabbits with TofR-6xHis and TofI-6xHis, respectively. Anti-TofR and anti-TofI antibodies were used to detect the corresponding proteins, and immunoreactive bands were visualized using ECL reagents (Bio-Rad) and captured using a ChemiDoc XRS+ (Bio-Rad). All data are representative of triplicate experiments.

## Purification of TofM-His

The pET-21b(+) expression system (Novagen) was used to express 6xHis-tagged TofM in *E. coli* BL21 (DE3). Overnight cultures (10 mL) of BL21 (DE3) harboring the expression plasmids were used to inoculate LB (1 L) containing the appropriate antibiotic. The culture was incubated with shaking (37°C, 200 rpm) until its $OD_{600}$ reached 1.4. At this point, the production of recombinant proteins was induced through the addition of 0.1 mM isopropyl β-d-1-thiogalactopyranoside. Then the induced culture was incubated overnight with shaking (20°C, 200 rpm). The cells were harvested through centrifugation

and the cell pellet was stored at −80°C until needed. The 6xHis-fusion TofM protein was purified using Ni-NTA Superflow resin (Qiagen), following the manufacturer's procedures.

## RNA electrophoretic mobility shift assay

The upstream regions of putative TofM target genes, including the T7 promoter at the 5′ end, were amplified using the primers shown in Table S3 in the supplemental material. The purified PCR products were used for *in vitro* RNA synthesis using the MAXIscript T7 kit (Invitrogen). The resulting RNAs were labeled with biotin using the Pierce™ RNA 3′ end biotinylation kit, as described by the manufacturer (Thermo Fisher Scientific). We used *katE1* as competitor RNAs, which were amplified using the PT7katE-F, PkatE-R, and MkatE-R primers, as shown in Fig. S3; Table S3 in the supplemental material. Purified TofM-His was incubated in binding buffer [10 mM HEPES (pH 7.3), 20 mM KCl, 1 mM $MgCl_2$, 1 mM DTT, and 1% (vol/vol) glycerol] containing 10 nM or 25 nM biotin-labeled RNA for 15 min at 28°C. For competition analyses, unlabeled target RNA at 100-fold molar excess was added to each reaction. Reaction mixtures were separated with non-denaturing 4% (wt/vol) polyacrylamide gels and transferred to nitrocellulose membranes, followed by the detection of relevant bands using streptavidin/horseradish peroxidase-derived chemiluminescence kits, as described by the manufacturer (Thermo Fisher Scientific). Images were obtained using the Chemi Doc XRS + instrument and the Image Lab Software (Bio-Rad). All data are representative of triplicate experiments.

## Measurement of amino acid levels

The *B. glumae* strains were subcultured by diluting the seed culture to an $OD_{600}$ of 0.05 and grown in LB broth at 37°C with shaking at 250 rpm. After 0, 2, 4, and 8 hours of subculture, CFUs were counted, and the culture supernatants were sampled by centrifugation and filtered through a 0.25-μm syringe filter. Amino acids remaining in the culture supernatants were quantified through high-performance liquid chromatography (HPLC, Ultimate 3000; Thermo Dionex) with an INNO C18 Column (5 μm, 4.6 mm × 150 mm; YoungJin Biochrom). Mobile phase A consisted of buffer (40 mM $Na_2HPO_4$), while mobile phase B consisted of distilled water, acetonitrile, and methanol (10:45:45, vol/vol/vol), flowing at a rate of 1.5 mL/min. The column temperature was maintained at 40°C, and the injection volume was 1 μL. The chromatography was generated by measuring the intensity of absorbance using both a fluorescence (FL) detector with an emission wavelength of 450 nm and excitation wavelength of 340 nm for OPA-derivatized samples, and an emission wavelength of 305 nm and excitation wavelength of 266 nm for FMOC-derivatized samples. In addition, a UV detector operating at 338 nm was used. To determine the concentration of amino acids, we used amino acid standards (Agilent 5061–3330 and Agilent 5062–2478). All experiments were independently replicated at least three times.

## Count for viable cells

Cells were inoculated into 2 mL of LB broth with appropriate antibiotics and were grown at 37°C and 250 rpm for 18 hours. Overnight cultures were then washed twice with fresh LB broth, and turbidity was adjusted to an $OD_{600}$ of 0.05 using a BioSpectrometer (Eppendorf). Next, 2 mL of subculture was added to glass test tubes (PYREX) for all assays. At the designated time point, we took 100 μL aliquots from each sample without washing, performed serial dilutions, and spread the diluted samples on LB agar medium. This allowed us to observe the formation of single CFUs and assess colony morphology. The LB agar plates were incubated at 37°C for 24 hours to allow colonies to grow. CFUs were counted under a dissecting microscope and multiplied by the appropriate dilution factor to calculate CFU/mL. The reported data represent the means of three biological replicates ± standard deviation.

## Identification of the mutations in spontaneous mutant strains

Chromosomal DNA was isolated from 20 spontaneous mutants cultured in LB broth using a method previously published by Sambrook et al. (39). To identify the mutation positions, we amplified the *tofI*, *tofR*, *qsmR*, and *obcAB* regions of 20 mutants using the primers (see Table S3 in the supplemental material) followed by direct sequencing.

## Monitoring the levels of oxalate and pH in the culture supernatant

Extracellular pH and oxalate levels were measured at an appropriate time point as described previously (38).

## Measurement of catalase activity

Catalase activity was assessed through a modified version of an assay described in a previous study (40). At designated time points, 1 mL of *B. glumae* cells was harvested after subculturing the seed culture, which was diluted to an $OD_{600}$ of 0.05, and grown in LB broth at 37°C with shaking at 250 rpm. The cells were washed twice with Tris-buffered saline [TBS, 50 mM Tris-Cl (pH 8.0), 150 mM NaCl] and resuspended in 100 μL TBS. The resulting bacterial suspension (100 μL) was placed in a 15 mL conical tube (SPL), followed by the addition of 100 μL of 1% Triton X-100 and 100 μL undiluted hydrogen peroxide (30%). After thoroughly mixing, the solutions were incubated at room temperature, and the height of the $O_2$-forming foam that remained present at 5 min was measured using a ruler. The relative catalase activity was determined by dividing the height of the $O_2$-forming foam of *B. glumae* strains by that of the wild type at each time point. The data represent the means of three biological replicates ± standard deviation.

## Measuring metabolic activity levels

Metabolic activities were quantified by measuring absorbance at 540 nm ($A_{540}$) for solvent-solubilized formazan. The principle of this method is that the redox potential of viable cells converts the water-soluble 3-(4,5-dimethylthiazol-2-yl)–2,5-diphenyltetrazolium bromide (MTT) into insoluble formazan (41). The seed culture and inoculation were prepared in the same manner when measuring catalase activity. The cells were washed twice with fresh LB and resuspended in 100 μL of LB. Then, 10 μL 12 mM MTT stock solution was added to each sample and the samples were incubated for 20 min at 37°C. Then the samples were centrifuged for 1 min, and 85 μL of supernatant was discarded. Subsequently, 200 μL DMSO was added to each sample and mixed thoroughly, followed by further incubation at 37°C for 10 min. Then $A_{540}$ was measured. The data represent the means of three biological replicates ±standard deviation.

## Statistical analysis

All experiments were conducted in triplicate with the respective controls. One-way analysis of variance (ANOVA) was used followed by Tukey's honest significant difference test as a post hoc analysis using SPSS statistical software (v. 26; IBM Corp.) to detect significant differences where required. A *P*-value of <0.05 was considered statistically significant.

## ACKNOWLEDGMENTS

This work was supported by the Basic Science Research Program through the National Research Foundation of Korea (NRF) funded by the Ministry of Education (2021R1I1A1A01040314).

## AUTHOR AFFILIATIONS

[1]Department of Agricultural Biotechnology, Seoul National University, Seoul, South Korea

²Research Institute of Agriculture and Life Sciences, Seoul National University, Seoul, South Korea

## AUTHOR ORCIDs

Eunhye Goo http://orcid.org/0000-0002-3247-5647
Ingyu Hwang http://orcid.org/0000-0002-4250-4165

## FUNDING

| Funder | Grant(s) | Author(s) |
|---|---|---|
| National Research Foundation of Korea (NRF) | 2021R1I1A1A01040314 | Eunhye Goo |

## AUTHOR CONTRIBUTIONS

Eunhye Goo, Conceptualization, Data curation, Formal analysis, Funding acquisition, Investigation, Resources, Validation, Visualization, Writing – original draft, Writing – review and editing | Ingyu Hwang, Conceptualization, Data curation, Formal analysis, Supervision, Writing – review and editing

## DATA AVAILABILITY

All data generated or analyzed during this study are included in this article and its supplemental files; further inquiries can be directed to the corresponding author.

## ADDITIONAL FILES

The following material is available online.

### Supplemental Material

**Supplemental Figures (Spectrum03353-23-s0001.pdf).** Figures S1 to S7.
**Supplemental Tables-references (Spectrum03353-23-s0002.pdf).** Tables S1 to S3.

### Open Peer Review

**PEER REVIEW HISTORY (review-history.pdf).** An accounting of the reviewer comments and feedback.

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
