## [Reviewer comments · Microbiology Spectrum]

Microbiology Spectrum

Control of bacterial quorum threshold for metabolic homeostasis and cooperativity

Eunhye Goo and Ingyu Hwang

Corresponding Author(s): Eunhye Goo, Seoul National University

Review Timeline:

Submission Date:	September 13, 2023
Editorial Decision:	October 10, 2023
Revision Received:	October 30, 2023
Accepted:	November 2, 2023

Editor: Giordano Rampioni

Reviewer(s): The reviewers have opted to remain anonymous.

Transaction Report:

DOI: <https://doi.org/10.1128/spectrum.03353-23>

October 10, 2023

Dr. Eunhye Goo
Seoul National University
Agricultural Biotechnology
5101 Bldg#200
Seoul 151-921
Korea, Republic of

Re: Spectrum03353-23 (Control of bacterial quorum threshold for metabolic homeostasis and cooperativity)

Dear Dr. Eunhye Goo:

Thank you for submitting your manuscript to Microbiology Spectrum. Your manuscript has been evaluated by two Reviewers with expertise in the area addressed in your study and it was the consensus view of these Reviewers that your paper contains interesting and solid data with significant potential impact. However, both Reviewers highlighted a number of issues that need to be addressed before manuscript acceptance. I will be glad to consider for publication in Microbiology Spectrum a revised version of your manuscript addressing all the criticisms raised by the Reviewers.

When submitting the revised version of your paper, please provide (1) point-by-point responses to the issues raised by the Reviewers as file type "Response to Reviewers," not in your cover letter, and (2) a PDF file that indicates the changes from the original submission (by highlighting or underlining the changes) as file type "Marked Up Manuscript - For Review Only". Please use this link to submit your revised manuscript - we strongly recommend that you submit your paper within the next 60 days or reach out to me. Detailed instructions on submitting your revised paper are below.

Link Not Available

Sincerely,

Giordano Rampioni

Journals Department
Reviewer comments:

Reviewer #1 (Comments for the Author):

Overview. The study by Goo and Hwang investigates quorum sensing in the rice pathogen *Burkholderia glumae*. A particular intriguing aspect of quorum sensing in this organism is the concept of "metabolic slowing" and anticipation of stress conditions at high cell density. The authors investigate the mechanism that determines the transition to the high-density state (quorum threshold), and the selection pressures that help maintain it. The study is methodically comprehensive and solid. I appreciate the

focused nature of the manuscript, but I find that it needs more detail in several places, as indicated below. In particular, certain experimental choices, designs, and conclusions drawn require more explanation, may need to be reconsidered, or presented differently. Text and figures should be edited in several places for clarity. Taken together, I believe the manuscript requires a major revision.

Main points:

1. Line 87, l150, and several other places: Please explain why qsmR mutants are considered "non-cooperative".
2. Line 97 on: CFU should be correlated with OD early on to appreciate when AHL samples were taken during the growth curve in Fig. 3A (sampling times are mentioned in the Materials, but should be mentioned earlier). In fact, there appears to be a surprising lack of correlation between CFU and OD that should be addressed. Between 8 and 10 hrs, CFU increases almost 10-fold, whereas OD only increases 2-fold (l306 and Fig. 3A)! Based on sampling time and OD, the densities chosen for LCD and HCD are remarkably close together.
3. Line 123: "predicted secondary structure...": How was this prediction made? Simply an observation "by eye", or was specific software used? There's no explanation in the Methods. Why is the hairpin structure considered to be "typical"?
4. Line 140 on: Cell densities in Fig. 3A should be reported on a log-scale. As shown, it appears that tofM mutation or AHL addition mainly affect the lag phase, whereas the growth rates in exponential phase are at least as high as in the WT.
5. Line 148: Higher levels of amino acids in the tofM mutant culture are consistent with metabolic slowing, but don't prove it. Any type of growth-inhibitory mutation (e.g. in cell division) would result in higher levels of substrate in the medium.
6. Line 155: Fig. 4 B and E: It is extremely surprising that mutations have almost identical trajectories under different conditions (tofM mutation vs AHL addition), with very little variation among replicates!
7. Line 158: The colony morphology and pH sensitivity of the QS mutants should be shown for comparison (as controls). The rationale for sequencing qsmR, obcB, and obcB genes should be made more clear.
8. Line 167: Please explain the significance of the experiments with tofI and tofRIM plasmids in the respective mutant backgrounds shown in Fig. 4F. Why were these strains/constructs used?
9. Line 175: Please provide more context and rationale why the focus is on oxalate and catalase production and how QsmR regulates these factors. What is the regulatory relationship between TofRI and QsmR?
10. Line 229: "more stringent QS threshold ... than reported for other bacteria", and l 241: "...but has been confirmed in only a few species." Can the authors cite studies that support these claims?
11. Line 247 and other places in the ms: The authors emphasize the importance of the QS threshold, yet have very little data that actually map and exactly define this threshold during culture growth, such as a time course of QS-dependent gene expression as a function of cell density. For example, Fig. 1 only has 3 CFUs at which AHL is measured; a reference to OD has already been suggested in point 2 above. At the least, the authors could report absolute QS-dependent catalase activity (rather than normalized, Fig.5) over time, and correlate with cell density.
12. Line 266: Fig. S7: This is an important figure that should be shown earlier in the results, as it raises an important question about the interpretation of the data. If qsmR expression isn't increased at LCD in the tofM mutant or in the presence of AHL, how can oxalate and catalase production be activated by this regulator? In other words, if qsmR expression is similar in the WT and tofM mutant, why would there be selection to inactivate qsmR in one but not the other?
13. Discussion in general: Can the authors summarize the mechanism for TofM function based on their findings? How is posttranscriptional regulation of tofI at the 5' UTR supposed to work?

Other:

1. Line 10: "the homoserine lactone (AHL)-mediated": replace "the" with "an"
2. Line 53: "Mechanisms" should be the singular "mechanism"
3. Lines 66-67: typo. Maybe supposed to be "how widespread QS modulation is among QS..."
4. Lines 69-70: It is stated that "studies" (plural) have shown something but then only one study is cited. Please correct.
5. Line 82-84: This sentence should be broken up to clearly distinguish two different perspectives here - requiring higher cell density in the presence of tofM, and producing high signal levels in the absence of tofM.
6. Line 96: The production of C6-HSL should be mentioned in the Intro. Is it a second AHL signal produced by *B. glumae*?
7. Line 99: "eye-detectable": Replace with "visible", or "detectable by eye"...
14. Line 101: Fig 1A: The labeling scheme is cumbersome for the reader. A brief descriptive label instead of numbers would help.
8. Line 117: "significant": consider a different term to avoid association with statistical significance, which wasn't quantified here.
9. Line 118: Fig 1C left panel: explain that the intense lower band is non-specific, to avoid confusion.
10. Line 130: It's not entirely clear why this fragment was used if binding to the overlapping fragment had already been shown. Instead, one could have used a fragment that does NOT include the binding site?
11. Line 155: Fig. 4: Two issues. First, the authors should make the color coding in the figure legend more clear. E.g. in panel B, BGR1 parent and qsmR mutant data refer to the same culture, whereas tofM and tofM qsmR refer to a DIFFERENT culture. Second, there are several issues with symbols and error bars. Panel E has the first 5 points of qsmR pink filled with red. All of the error bars for BGR1 are in pink so it looks like some points have more than 2 error bars (likely some need to be set to blue). It's odd that the pink error bars are behind the blue circles. Panel F shows an error bar in red at day 9.
12. Line 167: Why do all strains in Fig. 4E carry the "empty" vector pLAFR6?
13. Line 180: "no significant...pH...": It appears that stats was only done for panel A, not B in Fig S6B. Please clarify.
14. Line 183 on: The bar graphs in Fig. 5 are cumbersome to read; it might help to change to line graphs.
15. Line 236 panicle/spikelet: Please explain. Panicle of what plant species? Define spikelet for microbiologists.

16. Line 243: typo. Maybe supposed to be "...that remains unknown"
17. Line 266: Fig S7: Can the authors graphically indicate statistical significance in the figure?
18. Line 292-3: Insert "plasmid" before "carrying", and reword "marker exchanged into chromosome".
19. Reporting of stats in figures, especially Fig 3 B-D and Fig. 5: The statistics aren't well explained in the legend. What was compared? Was ANOVA done for all measurements at all time points together, or was a separate ANOVA done for each time point?

Reviewer #2 (Comments for the Author):

The authors did a wonderful job at characterizing the TofM-mediated repression of QS in *Burkholderia glumae*. They show, that the QS response will be activated at lower cell densities if TofM is inactivated. They also show, the mechanism of repression, which is post-transcriptional regulation of *tofl* messenger RNA. TofM binds to *tofl* mRNA and prevents its translation. Additionally, they show that an active QS will affect the central metabolism of the cells. The effect is so strong, that it drives the emergence of spontaneous QS receptor mutants. The experiments are controlled, clean, and clear/cut, and contain all the necessary controls. I believe that from an experimental standpoint, no additional experiments are needed. The only problem I can see in their manuscript is some interpretation of existing knowledge on the topic of Quorum sensing in the introduction and their interpretation of the results in comparison to other work in the following discussion, for which I suggest that a revision is needed. I suspect that the reason for this might be a language barrier.

The basic principle of the current "QS dogma" is that cells produce AHLs at a low basal rate; at high cell densities a lot of cells produce the signal and therefore the QS system is "activated". Because such accumulation dynamics correlate with cell density, and because at high cell densities, enough signal is produced to trigger a population-wide response, such communication is often believed to coordinate the cell behavior of high-density cells. The authors seem to have mixed cause and effect and believe that QS signal production is triggered by some other independent process at high cell density (at least this is what I infer from what they have written). This goes against the current consensus, however, where cell density merely correlates with AHL production (more cells, more AHL). The fact that the QS response will result (amongst other things) also in an increased production of the QS signal itself, thus generating a positive feedback loop, is probably additionally confounding. To better understand the issue I warmly suggest that the authors consult the perspective piece by Plat and Fuqua, 2010.

Additionally, to complicate things further; the QS signal (in this case an AHL) is not necessarily the only trigger for QS, and not necessarily all of the cells in a population might respond to QS (Smith and Schuster, PNAS, 2022), thus exhibiting phenotypic heterogeneity. While the authors have not measured the QS response at a single cell level, or identified any of the additional triggers that might derepress TofM, they could at least discuss/mention the latter. The above was elaborated more in detail in the recently published perspective piece by Spacapan et al., iScience 2023. The tight translational repression might not be as efficient, and sooner or later, during growth certain cells could become derepressed and start producing AHL. Indeed, the authors do observe a degree of AHL production in the wild type grown to high cell density. Because of the positive feedback loop, local AHL production should increase even more and trigger neighboring cells into an "ON" state, thus resulting in a most probably heterogeneously phenotypic pattern of activation.

Note, that the sensitivity of the *C. violaceum* overlay assay is very low, if one is not able to detect AHLs, this does not mean that there are none in the medium, it could be that it is only below the detection limit. Especially if the production of TofI is phenotypically heterogeneous. Additionally, assessing AHL quantity in such a manner will only yield an estimate for the production of the entire bacterial population.

If the authors correct for the main issue discussed then I would strongly suggest this work for publication.

Here are some more specific comments on the manuscript:

Line 13: Bacteria most likely don't determine the QS threshold, they just possess a mechanism that, in specific environments will result in QS activation and this will usually correlate with cell density.

Line 14 / 15: The QS response has been well described in many bacteria and I don't understand why making such a decision would constitute a problem.

Line 16/17/18: Why wouldn't TofM be able to bind to *tofl* mRNA at high cell densities? I would understand, however, that the very tight repression might not be 100% effective and that at high cell densities, some might nonetheless produce TofI and trigger a localized QS cascade due to the positive feedback loop.

Line 24: I do not understand how QS in this case maintains cooperativity if it forces "cheater" phenotype mutations. The premise of this conclusion is not clear enough.

Line 25: I do not understand what is meant by "social activity" and how the findings show diversification.

Line 27: Does this imply, that at HCD QS does not result in the same phenotype as at LCD?

Line 30-31: Cells use QS to coordinate phenotypes with cell density; as opposed to cell density triggering QS, see main point above.

Line 35-38: I agree that, if QS controls metabolic slowing, it is better for the cells to not activate it at low cell densities, however, there still has to be a reason probably, for why the QS-regulated phenotype is beneficial. Metabolic slowing could represent the trigger for a persisting state. I think that I understand what the authors mean and I agree with them, but it would be nice if they re-wrote the passage to convey that idea better.

Line 41: Cells monitor signal quantity which then correlates with cell density and not cell density directly. Reference 1, which is also cited, explains this well.

Line 45-47: At what conditions QS is activated? Due to the points mentioned above it does not only depend on cell density. See Plat and Fuqua, 2010 and Hense et al., 2007.

Line 48-52: They affect the dynamics of signal production and signal stability, therefore, depending on the specific environment, AHL concentration might not correlate with cell density in the same way.

Line 65: delaying instead of retarding

Line 66/67: Probably mean that QS modulation should be investigated more or something similar?

Line 68/69: Reference number 14 does not say that "quorum is reached at HCD", and reference 15 explains that cells ""sense"" AHL concentration which then only correlates with cell density at certain conditions.

Line 70: Reference 16 speaks about Pseudomonas growing on protein. While many similar examples exist it does not hold true for all bacteria, which might regulate different things with QS and live in different environments.

Line 75: I do not understand what is meant by cooperativity here, especially in the example of *B. glumae*. The premise should be elaborated.

Line 82: And most likely increases tofI expression.

Line 82-84: At high cell densities it just starts producing measurable quantities of AHL; this does not mean that the repression is 100% effective at LCD. One could then reason that the system has probably evolved to activate at very late growth stages.

Line 86 and afterward: I agree that the QS active metabolic phenotype will cause metabolic slowing, as the authors have shown this very nicely. However, saying it is important to avoid it at LCD implies that it does not cause metabolic slowing at HCD; so I find the point that the authors are trying to make here a bit confusing. The fact that QS does cause metabolic slowing is, however immensely interesting and does raise important questions in regards to the population growth dynamics.

Line 107: This indicated that tofM exerts a negative control on the average AHL production.

Line 150-153: You probably meant to say that you tested if the metabolic slowing of QS will select against QS in the fast-growing conditions. This is a very interesting result, that to me shows that in this case QS regulates a stress response phenotype and its implications are major; from the implications it has for "bet-hedging" to the fact that it shows QS regulation is irrelevant (and therefore better off repressed) in fast-growing conditions. Please try to re-phrase, as your results are very important for the field of QS, since the way it is written I have trouble understanding what your actual interpretation of the results may be.

Line 201: You added exogenous signal to the culture, but did not actually study intra-species cross-talk. I agree, however, that exogenous signals in nature might in some cases originate from other species.

Line 202: Are you implying that at high cell densities, no metabolic slowing is observed in your case?

Line 202-207: It has been known, since the discovery of QS in *Vibrio fischerii*, that QS regulation is subject to additional regulation besides the concentration of QS signal. (Soto-Aceves et al., 2023). Please notice, that also in this article the authors refer to the concentration of AHL when talking about "the quorum threshold" and not to cell density, which just happens to correlate with it.

Line 209-211: The authors of ref. 23 do not show the mechanism of action of RsaM, only that it derepresses AHL production. The mechanism might actually be similar to what the authors have found for TofM in *B. glumae*.

Line 213-214; As mentioned before, in ref 23 the authors did not specify that RsaM inhibits transcription of pfsI, they only show that the transcription of pfsI is increased; the mechanism of repression might still be post-translational as in the case of ToFM. Increased transcription might be due to the positive feedback loop of AHL QS systems.

Line 220: Unclear what the authors mean here

Line 226-228 At these densities measurable AHL quantities were produced and the system was activated. Please consider the positive feedback loop of AHL QS. It is most likely not cell density that directly triggered QS; as mentioned before.

Line 228-239: I fully agree with what I think the authors tried to convey here; that delaying virulence factor expression to late growth stages might significantly alter the pathogenicity dynamics of *B. glumae* and that this should be studied in a more pertinent "natural" environment; however I think that this paragraph should be re-written to better convey this intended meaning.

Line 240- 246: I don't understand what the authors meant to convey here.

Rest of the discussion: Please see the comment made for lines 150-153. Very often the authors raise rather general and non-specific questions and claim that these are understudied, however, it is very hard to evaluate this, because it is unclear to me what the specific question is in the first place. The authors show rather well that QS is active during late growth stages, and that QS causes metabolic slowing. The implications of these results are very interesting, however, the way the results are interpreted in the manuscript is confusing. What the authors clearly show in relation to the physiology of QS "active" cells is very important for the field of QS and merits clearer explanation.

Comments on materials and methods:

Line 286: Specify what is Affymetrix?

Line 286: The antibiotics were added only when needed I presume? Same goes for C8 AHL probably?

Line 304-305 So the initial inoculum was 20x diluted culture of OD600 0.05? Is this true for all subsequent experiments?

Line 308: In the given reference there is no info on AHL extraction or *C.violaceum* assays.

Line 363-364; You must have centrifuged the culture before filtering it?

Line 370: What was measured to produce the chromatography when separating the amino acids on the column? How was the concentration of amino acid determined?

372-281: Does this part just intend to say that you took 2 ml aliquots from growing cultures (and not from washed overnight "seed cultures), serially diluted them, and plated them? How does this assess viability of cells? Was the "seed" overnight culture washed also in the other experiments?

Line 385 can you specify the page of the manual referenced as 31?

Line 393-395 The cells were subcultured again after taking an aliquot? Probably you meant to describe the conditions as in lines 408-410?

Line 408-410 I assume that in every instance the seed culture and inoculation were prepared as is specified here. Please correct the other instances if this is so or maybe mention once how bacteria were cultured and that only an aliquot was taken for each analysis when needed.

Line 419: By "honestly significant difference post hoc analysis" I presume the authors meant the Tukey "honest significant difference test"? Additionally, the p-value threshold of 0.05 does not mean the result was significant, it only indicates the false positive rate. The authors, however, might have chosen this threshold to be considered significant.

Overall comment:

The work done here is novel, impactful, and important for the field of QS; and I strongly recommend it for publication. However, the authors probably struggle with a language barrier, which I strongly feel should not invalidate their work. I also hope that the review did not come across as demeaning; this was not my intention. I am sure the authors understand their research very well, it's just that the way they expressed their results is confusing in places and I suggest to clarify.

References:

Platt TG, Fuqua C. What's in a name? The semantics of quorum sensing. *Trends Microbiol.* 2010 Sep;18(9):383-7. doi: 10.1016/j.tim.2010.05.003. Epub 2010 Jun 21. PMID: 20573513; PMCID: PMC2932771.

Spacapan, M., Bez, C., & Venturi, V. (2023). Quorum sensing going wild. *iScience.*

Hense, B., Kuttler, C., Müller, J. et al. Does efficiency sensing unify diffusion and quorum sensing?. *Nat Rev Microbiol* 5, 230-239 (2007). <https://doi.org/10.1038/nrmicro1600>

Soto-Aceves, M. P., Diggle, S. P., & Greenberg, E. P. (2023). Microbial Primer: LuxR-LuxI Quorum Sensing. *Microbiology*, 169(9), 001343.

Smith, P., and Schuster, M. (2022). Antiactivators prevent self-sensing in *Pseudomonas aeruginosa* quorum sensing. *Proc. Natl. Acad. Sci. USA* 119, 2201242119.

Staff Comments:

Preparing Revision Guidelines

Please return the manuscript within 60 days; if you cannot complete the modification within this time period, please contact me. If you do not wish to modify the manuscript and prefer to submit it to another journal, please notify me of your decision immediately so that the manuscript may be formally withdrawn from consideration by Microbiology Spectrum.

Reviewer comments:

Reviewer #1 (Comments for the Author):

Overview. *The study by Goo and Hwang investigates quorum sensing in the rice pathogen Burkholderia glumae. A particular intriguing aspect of quorum sensing in this organism is the concept of "metabolic slowing" and anticipation of stress conditions at high cell density. The authors investigate the mechanism that determines the transition to the high-density state (quorum threshold), and the selection pressures that help maintain it. The study is methodically comprehensive and solid. I appreciate the focused nature of the manuscript, but I find that it needs more detail in several places, as indicated below. In particular, certain experimental choices, designs, and conclusions drawn require more explanation, may need to be reconsidered, or presented differently. Text and figures should be edited in several places for clarity. Taken together, I believe the manuscript requires a major revision.*

Main points:

1. Line 87, I150, and several other places: Please explain why qsmR mutants are considered "non-cooperative".

- Mutation of the *qsmR* gene results in impaired cooperative activities, such as oxalate biosynthesis. In *Burkholderia glumae*, quorum sensing serves as a mechanism to anticipate stationary phase or life at the carrying capacity of a population by activating the expression of cytoplasmic enzymes, altering cellular metabolism, and producing a shared resource or public good (in this case, oxalate). To address the reviewer's concern, we have included additional explanations on lines 85–88 in the revised manuscript.

2. Line 97 on: CFU should be correlated with OD early on to appreciate when AHL samples were taken during the growth curve in Fig. 3A (sampling times are mentioned in the Materials, but should be mentioned earlier). In fact, there appears to be a surprising lack of correlation between CFU and OD that should be addressed. Between 8 and 10 hrs, CFU increases almost 10-fold, whereas OD only increases 2-fold (I306 and Fig. 3A)! Based on sampling time and OD, the densities chosen for LCD and HCD are remarkably close together.

- The optical density (OD) and colony forming units (CFU) can indeed be related, but it is important to note that they can also differ depending on the culture conditions, such as aeration and culture volume. We monitored OD in real-time using a microplate reader while bacterial cells were growing in 200 μ L of LB broth in each well of a microplate with shaking at 250 rpm. The results of OD monitoring were primarily used to illustrate the differences between *B. glumae* strains in the early growth stage. However, for all other assays, including the determination of CFU, *B. glumae* cells were cultured in 2 mL of LB broth in 16

mm x 125 mm, 16.5-mL glass test tubes (PYREX) with shaking at 250 rpm. To address this issue and avoid potential confusion, we have now included information about the specific culture volumes on lines 336–339 of the Materials section in the revised manuscript. This will help readers better understand the differences in culture conditions and their impact on the measurements.

3. Line 123: "predicted secondary structure...": How was this prediction made? Simply an observation "by eye", or was specific software used? There's no explanation in the Methods. Why is the hairpin structure considered to be "typical"?

- We predicted the secondary structure of the promoter region of *tofI* using MXfold software at web server (<http://www.dna.bio.keio.ac.jp/mxfold/>). This software is based on the source code of CONTRAfold and is released under the MIT license. We have included an explanation regarding the methodology for secondary structure prediction in the figure legend. In addition, we have removed the word "typical" to avoid any potential misinterpretation.

4. Line 140 on: Cell densities in Fig. 3A should be reported on a log-scale. As shown, it appears that *tofM* mutation or AHL addition mainly affect the lag phase, whereas the growth rates in exponential phase are at least as high as in the WT.

- As mentioned previously, we believe that measuring optical density provides a more detailed representation of the initial growth difference between strains, even though the *tofM* mutation or addition of homoserine lactone (AHL) primarily affects the lag phase. While we acknowledge that the suggestion to report cell densities on a log-scale is valuable, it is important to note that using OD results in Fig. 3A for a direct comparison with sampling times in other phenotypic studies could be misleading.

Indeed, our results indicated that the *tofM* mutation and addition of AHL primarily affected the lag phase of growth, leading to a delay in the onset of exponential growth. However, noted by the reviewer, once the cells enter the exponential phase, their growth rates become comparable to the wild type (WT). This observation emphasizes the importance of the lag phase in understanding the impact of the *tofM* mutation and addition of AHL on growth dynamics, as it results in a delay in the initiation of active growth.

5. Line 148: Higher levels of amino acids in the *tofM* mutant culture are consistent with metabolic slowing, but don't prove it. Any type of growth-inhibitory mutation (e.g. in cell division) would result in higher levels of substrate in the medium.

- While it is true that this observation alone does not constitute definitive proof, we did not observe any growth-inhibitory mutations, such as those affecting cell division, in the *tofM* mutant under the microscope (data not shown). Moreover, our conclusion regarding the reduced growth rate of the *tofM* mutant being attributable to advanced metabolic slowing was grounded in the existing literature. It has been reported (ref. 5) that quorum sensing acts as a metabolic brake to ensure metabolic homeostasis of primary metabolism. Given this context, we inferred that the higher amino acid levels are a result of metabolic slowing in the *tofM* mutant. We have added this explanation on lines 153–154 in the revised manuscript to

clarify our reasoning and strengthen the scientific basis of our conclusion.

6. Line 155: Fig. 4 B and E: It is extremely surprising that mutations have almost identical trajectories under different conditions (tofM mutation vs AHL addition), with very little variation among replicates!

- Independent repeat experiments of monitoring CFUs from the *tofM* mutant and with the addition of AHL consistently showed almost identical trajectories. This robust consistency among replicates provided strong support for our conclusions.

7. Line 158: The colony morphology and pH sensitivity of the QS mutants should be shown for comparison (as controls). The rationale for sequencing *qsmR*, *obcA*, and *obcB* genes should be made more clear.

- Results of colony morphology (ref. 26) and pH sensitivity (ref. 23) in quorum sensing (QS) mutants have been reported previously, and were omitted from this manuscript to avoid redundancy and make the presentation more concise. To enhance the clarity of our rationale for sequencing *qsmR*, *obcA*, and *obcB* genes, we have provided additional explanations on lines 171–176 in the revised manuscript.

8. Line 167: Please explain the significance of the experiments with *tofI* and *tofRIM* plasmids in the respective mutant backgrounds shown in Fig. 4F. Why were these strains/constructs used?

- We conducted experiments using the *tofI* and *tofRIM* plasmids to demonstrate the impact of a higher copy number of *TofI* on the appearance of spontaneous mutants for *qsmR*. This experimental approach further reinforced our main finding concerning translational control of the QS signal synthase gene in *B. glumae*. The introduction of the *tofI* plasmid illustrated the effect of increased *TofI* level, which contributed to the observed outcomes. In addition, use of the *tofRIM* plasmid served as a control for the *tofI* plasmid, helping to differentiate the effects of *TofI* from other factors in the mutant backgrounds.

9. Line 175: Please provide more context and rationale why the focus is on oxalate and catalase production and how *QsmR* regulates these factors. What is the regulatory relationship between *TofRI* and *QsmR*?

- The complex of C8-HSL and *TofR* activates *qsmR* gene expression, subsequently regulating various social behaviors in *B. glumae*. Our focus on oxalate and catalase production was rooted in their crucial roles in the survival and adaptation of *B. glumae*. Specifically, *QsmR* plays a direct role in the regulation of these factors. The *obcA* and *obcB* genes, encoding oxalate biosynthetic proteins, are directly regulated by *QsmR* (ref. 23). Oxalate production is essential for survival of the bacterium in amino acid-based media after stationary phase, as it helps neutralize the alkaline toxicity resulting from amino acid deamination (ref. 23). In addition, *QsmR* also directly activates expression of the *katG* gene, which is responsible for responding to oxidative stress in *B. glumae* (ref. 22). To provide a clearer rationale for focusing on *QsmR*-controlled oxalate and catalase production, we have included additional information on lines 193–196 in the revised manuscript, explaining that “These factors are essential for survival and adaptation of *B.*

glumae; where biosynthesized oxalate neutralizes the alkaline toxicity resulting from deamination in amino acid-based media after the stationary phase, and QsmR directly activates the *katG* gene, which responds to oxidative stress in *B. glumae*.”.

10. Line 229: "more stringent QS threshold ... than reported for other bacteria", and I 241: "...but has been confirmed in only a few species." Can the authors cite studies that support these claims?

- The assertion that our study reveals a more stringent QS threshold compared to other bacteria was based on previous research primarily focusing on well-studied QS bacteria, such as *Vibrio* and *Pseudomonas* species. While these organisms have been examined extensively, the specific threshold values have not been universally determined for all bacterial species. Our study, therefore, adds value by providing an additional example of the regulation of the QS threshold in *B. glumae*.

11. Line 247 and other places in the ms: The authors emphasize the importance of the QS threshold, yet have very little data that actually map and exactly define this threshold during culture growth, such as a time course of QS-dependent gene expression as a function of cell density. For example, Fig. 1 only has 3 CFUs at which AHL is measured; a reference to OD has already been suggested in point 2 above. At the least, the authors could report absolute QS-dependent catalase activity (rather than normalized, Fig.5) over time, and correlate with cell density.

- In a previous study (ref. 23), we reported the results of QS-dependent RNA-seq analysis before and after onset of QS. We found that the QS threshold, defined as the cell density at which QS-dependent genes begin to be expressed, is higher in *B. glumae* than that in other well-studied QS bacteria, such as *Vibrio* and *Pseudomonas* species. At the time, we could not provide a detailed explanation for why QS turns off at a specific cell density of 3×10^8 CFU/mL, unlike other QS bacteria where QS is typically activated. In the present study, our primary objective was to determine the regulatory mechanisms of the QS threshold and its biological significance. The presentation of normalized catalase activity in Fig. 5 was not intended to show QS-dependent enzyme activity correlated with cell density, but rather to provide an example of selection pressure on the *qsmR* gene in the presence of the *tofM* mutation or with addition of AHL during growth. We recognize that more comprehensive time course data and correlations with cell density would provide valuable insights, and we appreciate the suggestion. However, due to the focus and scope of this study, our objective was to elucidate the regulatory mechanisms underlying the QS threshold, as stated in the manuscript.

12. Line 266: Fig. S7: This is an important figure that should be shown earlier in the results, as it raises an important question about the interpretation of the data. If *qsmR* expression isn't increased at LCD in the *tofM* mutant or in the presence of AHL, how can oxalate and catalase

production be activated by this regulator? In other words, if *qsmR* expression is similar in the WT and *tofM* mutant, why would there be selection to inactivate *qsmR* in one but not the other?

- We acknowledge the importance of Fig. S7, and we appreciate the reviewer's attention to this aspect of our study. The placement of this figure in the Supplementary Materials was a deliberate decision to maintain the clarity of interpretation of the main results presented in the Results section. The primary focus of the qRT-PCR experiments, as described in the manuscript, was to determine whether QS is advanced by *tofM* mutation or addition of AHL, with a specific emphasis on the timing and regulatory aspects of this process. Based on our observations, the spontaneous mutation in the *qsmR* gene occurred 5 days after subculture, likely due to prolonged exposure to continuous metabolic stress, including metabolic and oxidative stress. Extraction of RNA for qRT-PCR to quantify the expression of *obcAB*, *qsmR*, and *flhD* was conducted at 8 and 10 h after subculture, which was within the growth period during which the QS-related phenotypes were investigated. In addition, it should be noted that the expression of *obcAB* genes can also be influenced by factors such as pH and possibly other as yet unidentified mechanisms. QsmR is not the sole factor responsible for activating these genes. Furthermore, catalase activity is governed by multiple genes, including *katG*. QsmR only regulates the expression of *katG* at the transcriptional level. To prevent any potential confusion in the interpretation of the results of our study, we have included the qRT-PCR results of *obcAB*, *qsmR*, and *flhD* at low cell density (LCD) and high cell density (HCD) in both the wild type and *tofM* mutant in the Supplemental Materials as Fig. S7. This placement allows for a more detailed examination of the specific qRT-PCR data, while maintaining the clarity and focus of the main findings presented in the Results section.

13. Discussion in general: Can the authors summarize the mechanism for TofM function based on their findings? How is posttranscriptional regulation of *tofI* at the 5' UTR supposed to work?

- Our findings showed that TofM functions by binding to the *tofI* mRNA to prevent translation of QS signal synthase TofI at low cell density. When the cell density reaches a critical threshold and C8-HSL accumulates, the complex of C8-HSL and TofR activates the transcription of the *tofI* gene through an autoinduction mechanism. Consequently, the copy number of *tofI* mRNA becomes higher than that of the TofM, leading to the activation of QS via a positive feedback loop in *B. glumae*. We have included these sentences on lines 265–269 in the revised manuscript.

Other:

1. Line 10: "the homoserine lactone (AHL)-mediated": replace "the" with "an"

- Amended accordingly.

2. Line 53 "Mechanisms" should be the singular "mechanism"

- Amended accordingly.

3. Lines 66-67: typo. Maybe supposed to be "how widespread QS modulation is among QS..."

- Amended accordingly.

4. Lines 69-70: It is stated that "studies" (plural) have shown something but then only one study is cited. Please correct.

- We have added a reference.

5. Line 82-84: This sentence should be broken up to clearly distinguish two different perspectives here - requiring higher cell density in the presence of *tofM*, and producing high signal levels in the absence of *tofM*.

- We have split the sentence into two parts. First, we were interested in why *B. glumae* requires higher cell density than other QS bacteria to meet the QS threshold. Second, our aim was to determine the molecular mechanisms underlying the abnormally high level of C8-HSL production in the *tofM* mutant, as reported previously (ref. 25)."

6. Line 96: The production of C6-HSL should be mentioned in the Intro. Is it a second AHL signal produced by *B. glumae*?

- We have briefly mentioned the production of C6-HSL in the Introduction in the revised manuscript. *B. glumae* is known to produce multiple AHL signals, including C6-HSL and C8-HSL. However, it should be noted that among these AHL signals, only C8-HSL binds to TofR, the AHL signal receptor, and plays a pivotal role in controlling density-dependent behaviors in *B. glumae*.

7. Line 99: "eye-detectable": Replace with "visible", or "detectable by eye"...

- We have changed the term to "detectable by eye" and "visible" in the revised manuscript.

14. Line 101: Fig 1A: The labeling scheme is cumbersome for the reader. A brief descriptive label instead of numbers would help.

- We have added a brief descriptive label in Figure 1A in the revised manuscript.

8. Line 117: "significant": consider a different term to avoid association with statistical significance, which wasn't quantified here.

- We have removed the word "significant" on line 128–129 in the revised manuscript.

9. Line 118: Fig 1C left panel: explain that the intense lower band is non-specific, to avoid confusion.

- We have added an explanation on line 130 to clarify that the intense lower band observed in the left panel of Fig. 1C is nonspecific.

10. Line 130: It's not entirely clear why this fragment was used if binding to the overlapping fragment had already been shown. Instead, one could have used a fragment that does NOT include the binding site?

- Our choice to use the 306-nt long *tofI* mRNA fragment was based on the logical progression of our experiments. Initially, the results of our first electrophoretic mobility shift assay (EMSA) demonstrated

binding to the full-length 763-nt *tofI* mRNA fragment. Subsequently, we performed a more focused and detailed analysis by identifying the specific target binding site of TofM.

11. Line 155: Fig. 4: Two issues. First, the authors should make the color coding in the figure legend more clear. E.g. in panel B, BGR1 parent and *qsmR* mutant data refer to the same culture, whereas *tofM* and *tofM qsmR* refer to a DIFFERENT culture. Second, there are several issues with symbols and error bars. Panel E has the first 5 points of *qsmR* pink filled with red. All of the error bars for BGR1 are in pink so it looks like some points have more than 2 error bars (likely some need to be set to blue). It's odd that the pink error bars are behind the blue circles. Panel F shows an error bar in red at day 9.

- We have revised the figure legend in accordance with the reviewer's comments. (B) Growth of BGR1 parent (black circle) and *tofM::EZ-Tn5* parent (red triangle outline) was monitored in LB broth over 10 days of closed batch culture. In addition, we also monitored the growth of the *qsmR* mutant (black dot) and *tofM/qsmR* mutant (solid red triangle), which derived from the BGR1 parent and *tofM::EZ-Tn5* parent, respectively. We have made edits to the first 5 points of *qsmR* in Panel E, which are now filled in pink. In addition, we have addressed the error bar issues in Figure 4.

12. Line 167: Why do all strains in Fig. 4E carry the "empty" vector pLAFR6?

- All strains shown in Fig. 4E carried an empty vector as controls for comparison with the results presented in Fig. 4F.

13. Line 180: "no significant...pH...": It appears that stats was only done for panel A, not B in Fig S6B. Please clarify.

- The absence of statistical analysis of the results in Fig. S6B was a deliberate choice based on previous findings (ref. 23). These results did not reveal biologically significant environmental pH differences between WT and *tofM* mutant, even though there were statistical differences. Therefore, we did not present the statistics for Fig. S6B to avoid potentially misleading interpretations.

14. Line 183 on: The bar graphs in Fig. 5 are cumbersome to read; it might help to change to line graphs.

- We initially used line graphs in Fig. 5 but intentionally switched to bar graphs to better emphasize the differences relative to the WT strain, and thus enhance clarity and highlight variations in results across different experimental conditions.

15. Line 236 panicle/spikelet: Please explain. Panicle of what plant species? Define spikelet for microbiologists.

- The term panicle in our study refers to the terminal component of the rice tiller, and is specific to rice (*Oryza sativa*). A spikelet is the basic unit of the panicle in rice and is commonly used in plant biology terminology. We have included this clarification in the manuscript on line 283–285 to ensure that microbiologists and other readers have a clear understanding of these plant-related terms in the context of

our study.

16. Line 243: typo. Maybe supposed to be "...that remains unknown"

- Amended accordingly.

17. Line 266: Fig S7: Can the authors graphically indicate statistical significance in the figure?

Graphically indicating statistical significance in Fig. S7 is challenging due to the variation in *P*-values among the samples, except for *obcB* at low cell density (LCD) and *flhD* at high cell density (HCD). This variability makes it difficult to apply a uniform graphical representation of significance across the entire figure.

18. Line 292-3: Insert "plasmid" before "carrying", and reword "marker exchanged into chromosome".

- Amended accordingly.

19. Reporting of stats in figures, especially Fig 3 B-D and Fig. 5: The statistics aren't well explained in the legend. What was compared? Was ANOVA done for all measurements at all time points together, or was a separate ANOVA done for each time point?

- A separate ANOVA was conducted for each time point in the figures, rather than a single ANOVA covering all measurements at all time points. We have improved the explanations in the legends of Fig. 3 B–D, Fig. 5, Fig. S4, Fig. S5, and Fig. S6 to specify which comparisons were made and the statistical methods used.

Reviewer #2 (Comments for the Author):

The authors did a wonderful job at characterizing the TofM-mediated repression of QS in Burkholderia glumae. They show, that the QS response will be activated at lower cell densities if TofM is inactivated. They also show, the mechanism of repression, which is post-transcriptional regulation of tofI messenger RNA. TofM binds to tofI mRNA and prevents its translation. Additionally, they show that an active QS will affect the central metabolism of the cells. The effect is so strong, that it drives the emergence of spontaneous QS receptor mutants. The experiments are controlled, clean, and clear/cut, and contain all the necessary controls. I believe that from an experimental standpoint, no additional experiments are needed. The only problem I can see in their manuscript is some interpretation of existing knowledge on the topic of Quorum sensing in the introduction and their interpretation of the results in comparison to other work in the following discussion, for which I suggest that a revision is needed. I suspect that the reason for this might be a language barrier.

The basic principle of the current "QS dogma" is that cells produce AHLs at a low basal rate; at high cell densities a lot of cells produce the signal and therefore the QS system is "activated". Because such accumulation dynamics correlate with cell density, and because at high cell densities, enough signal is produced to trigger a population-wide response, such communication is often believed to coordinate the cell behavior of high-density cells. The authors seem to have mixed cause and effect and believe that QS signal production is triggered by some other independent process at high cell density (at least this is what I infer from what they have written). This goes against the current consensus, however, where cell density merely correlates with AHL production (more cells, more AHL). The fact that the QS response will result (amongst other things) also in an increased production of the QS signal itself, thus generating a positive feedback loop, is probably additionally confounding. To better understand the issue I warmly suggest that the authors consult the perspective piece by Plat and Fuqua, 2010.

- Our intention was not to suggest that QS signal production is triggered by another independent process at high cell density. We understand that *TofM* modulates QS signal production until it generates a positive feedback loop at high cell density. Our results showed that the *tofM* mutation or addition of AHL to cells at low density, distinct from cells at high density, leads to slow growth in the early growth stage, metabolic imbalance, and the emergence of noncooperative spontaneous *qsmR* mutants. We also acknowledge that our findings challenge the current consensus where cell density is seen as merely correlated with AHL production.

To provide context, it is essential to note that the starting point for our study was based on density issues in *Burkholderia glumae*. The threshold for QS onset in *B. glumae* was found to be significantly higher compared to other QS bacteria. For example, induction of QS-dependent bioluminescence occurred in *Vibrio fischeri* at OD₆₀₀ of 0.2–0.4, equivalent to 1×10^8 CFU/mL (ref. 24). In *Pseudomonas aeruginosa*, expression levels of the QS-dependent *lasB* gene can be measured by GFP fluorescence levels of *lasB*::GFP from the cells at OD₆₀₀ = 0.2 (ref. 13). By contrast, the QS onset density for *B. glumae* under laboratory conditions was OD₆₀₀ = 1.5, equivalent to 1×10^9 CFU/mL, which is markedly higher, and the expression of QS-regulated genes was not detected until reaching the QS threshold (ref. 23). This difference raises intriguing questions about the underlying mechanisms and ecological implications of QS signaling in different bacteria. Our research was performed to elucidate these aspects, considering the potential consequences of artificially elevated concentrations of QS signals or early QS activation, which can occur in natural environments. We have revised the manuscript to provide a clearer explanation.

Additionally, to complicate things further; the QS signal (in this case an AHL) is not necessarily the only trigger for QS, and not necessarily all of the cells in a population might respond to QS

(Smith and Schuster, PNAS, 2022), thus exhibiting phenotypic heterogeneity. While the authors have not measured the QS response at a single cell level, or identified any of the additional triggers that might derepress TofM, they could at least discuss/mention the latter. The above was elaborated more in detail in the recently published perspective piece by Spacapan et al., iScience 2023. The tight translational repression might not be as efficient, and sooner or later, during growth certain cells could become derepressed and start producing AHL. Indeed, the authors do observe a degree of AHL production in the wild type grown to high cell density. Because of the positive feedback loop, local AHL production should increase even more and trigger neighboring cells into an "ON" state, thus resulting in a most probably heterogeneously phenotypic pattern of activation.

- We conducted our experiments under controlled laboratory conditions involving vigorous shaking in test tubes and using complex media, such as LB. We acknowledge that in natural environments, the ecological context of QS regulation is complex and influenced by various aspects of the surroundings. However, the primary focus of our study was to investigate the impact of TofM on QS signal production, not single-cell level or other triggers for TofM derepression. We assumed that TofM affects QS signal production until a positive feedback loop is generated at high cell density. We appreciate the reviewer's valuable insights and have revised our manuscript to include a deeper discussion on recent topics such as 'bacterial civilizations' and related emerging issues in the field.

Note, that the sensitivity of the C. violaceum overlay assay is very low, if one is not able to detect AHLs, this does not mean that there are none in the medium, it could be that it is only below the detection limit. Especially if the production of TofI is phenotypically heterogeneous. Additionally, assessing AHL quantity in such a manner will only yield an estimate for the production of the entire bacterial population.

- While we acknowledge that the sensitivity of the *C. violaceum* overlay assay may have limitations and could miss AHLs present at concentrations below the limit of detection, our primary goal was to demonstrate the presence of AHLs, specifically C8-HSL and C6-HSL, in *B. glumae* under the culture conditions we employed (LB broth and shaking at 250 rpm). We hypothesized that, under these conditions, there was no apparent phenotypic heterogeneity in AHL production and QS response within our bacterial population.

If the authors correct for the main issue discussed then I would strongly suggest this work for publication.

Here are some more specific comments on the manuscript:

Line 13: Bacteria most likely don't determine the QS threshold, they just possess a mechanism that, in specific environments will result in QS activation and this will usually correlate with cell density.

- We appreciate your perspective on the mechanism of QS threshold determination, and you are correct in highlighting that bacteria typically possess a mechanism that, under specific environmental conditions, results in QS activation, often correlated with cell density. Nevertheless, the precise mechanism by which QS bacteria respond to environmental cues leading to quorum threshold determination remains an area for further study. Little is known about the intricate interplay between bacterial physiology and social traits in the context of quorum threshold regulation.

Line 14 / 15: The QS response has been well described in many bacteria and I don't understand why making such a decision would constitute a problem.

- We used the term "problem" to describe the specific biological challenges stemming from exposure to high levels of QS signals, which can impact bacterial physiological and social traits. Our study was focused on investigating these intricate dynamics, and we have provided evidence to support our findings. We will ensure that the manuscript provides a clear and precise explanation of this concept to prevent any misunderstanding.

Line 16/17/18: Why wouldn't TofM be able to bind to tofI mRNA at high cell densities? I would understand, however, that the very tight repression might not be 100% effective and that at high cell densities, some might nonetheless produce TofI and trigger a localized QS cascade due to the positive feedback loop.

- We agree that TofM does not completely inhibit its binding to *tofI* mRNA at high cell densities, allowing for TofI production and potential localized QS cascades due to the positive feedback loop. In accordance with your advice, we have relocated "at low cell density (LCD)" to after "prevents QS signal biosynthesis" in the revised manuscript. As noted by reviewer 1, we have summarized the mechanism for the function of TofM based on our findings in the Discussion section.

Line 24: I do not understand how QS in this case maintains cooperativity if it forces "cheater" phenotype mutations. The premise of this conclusion is not clear enough.

- We appreciate the reviewer's comment and the need for clarification regarding the maintenance of cooperativity in the context of QS. While the concept of "cheaters" has been discussed in the literature (ref. 26), it is important to note that the spontaneous *qsmR* mutant in our study does not fit the definition of a "cheater", as observed in other bacteria such as *Pseudomonas aeruginosa*, as it did not appear under conditions requiring costly public goods. We also wish to clarify that the spontaneous *qsmR* mutant has a genetic mutation and not phenotypic mutation. The bacterial genetic system has evolved to maintain social activity among QS bacteria while addressing the challenges posed by environments where signals

generated by natural neighbors exceed their QS threshold. This perspective highlights the complexity of bacterial behavior and their strategies to balance social cooperation with adaptive responses to varying environmental conditions.

Line 25: I do not understand what is meant by "social activity" and how the findings show diversification.

- In this context, "social activity" refers to the coordinated behaviors and interactions that QS bacteria exhibit as a result of their density-dependent communication. Our findings demonstrate diversification in the bacterial genetic system by revealing the mechanisms that enable bacteria to adapt to changing environmental conditions and challenges presented when signals generated by their natural neighbors exceed their QS threshold.

Line 27: Does this imply, that at HCD QS does not result in the same phenotype as at LCD?

- This does not imply that QS results in the different phenotype at LCD and HCD. Our findings indicated that introducing the *tofM* mutation or adding AHL to cells at low density, which differs from cells at high density, results in slow growth during the early growth phase, metabolic disorder, and the appearance of noncooperative, spontaneous *qsmR* mutants.

Line 30-31: Cells use QS to coordinate phenotypes with cell density; as opposed to cell density triggering QS, see main point above.

- Our intention was not to suggest that QS signal production is initiated by a distinct independent process at high cell density, nor do we suggest that QS is exclusively triggered by cell density. Nonetheless, the outcomes observed when cells at low density are exposed to high levels of QS signals differ from the gradual accumulation dynamics typically associated with cell density.

Line 35-38: I agree that, if QS controls metabolic slowing, it is better for the cells to not activate it at low cell densities, however, there still has to be a reason probably, for why the QS-regulated phenotype is beneficial. Metabolic slowing could represent the trigger for a persisting state. I think that I understand what the authors mean and I agree with them, but it would be nice if they re-wrote the passage to convey that idea better.

- The QS-regulated phenotype, including metabolic slowing, indeed represents a mechanism for cells to persist in a state that optimizes their adaptation to specific conditions. Lack of stringent control serves as a selective force, leading to the emergence of noncooperative cells. This highlights the vital role of precise QS modulation in maintaining the integrity of the bacterial QS system. We have revised the passage for clarity.

Line 41: Cells monitor signal quantity which then correlates with cell density and not cell density directly. Reference 1, which is also cited, explains this well.

- We have amended the first line of the Introduction and cited reference 1 in the revised manuscript.

Line 45-47: At what conditions QS is activated? Due to the points mentioned above it does not only depend on cell density. See Plat and Fuqua, 2010 and Hense et al., 2007.

- QS activation depends on the cell density threshold and the presence of a sufficient amount of the signal molecule, which is influenced by various biotic and abiotic factors. It is a complex process that involves the accumulation of the signaling molecule to concentrations at which cells can perceive and respond to it. In the revised manuscript, we have provided a broader perspective on QS regulation, acknowledging the significant role of environmental factors in shaping its ecological and adaptive implications.

Line 48-52: They affect the dynamics of signal production and signal stability, therefore, depending on the specific environment, AHL concentration might not correlate with cell density in the same way.

- We have revised the sentences. "Various factors, including temperature, pH, nutrient availability, and the presence of signal molecules released by QS bacteria in the environment, can influence the QS threshold (1-3, 6). In addition, the presence of receptors or transcriptional regulators that bind to signaling molecules can impact the QS threshold (7-13). It is important to note that the mechanisms governing QS threshold determination may exhibit variability among bacterial species and ecological niches."

Line 65: delaying instead of retarding

- Amended accordingly.

Line 66/67: Probably mean that QS modulation should be investigated more or something similar?

- The extent and variations of QS modulation across different bacterial species and environmental conditions are intriguing subjects for future research.

Line 68/69: Reference number 14 does not say that "quorum is reached at HCD", and reference 15 explains that cells ""sense"" AHL concentration which then only correlates with cell density at certain conditions.

- We have moved reference 14 to just before the comma, and amended the sentence after the comma in the revised manuscript. "the QS threshold is generally reached at high cell densities although cells sense AHL concentration, which is correlated with cell density under certain conditions (ref. 15)."

Line 70: Reference 16 speaks about Pseudomonas growing on protein. While many similar examples exist it does not hold true for all bacteria, which might regulate different things with QS and live in different environments.

- Reference 16 indeed highlights the specific case of *Pseudomonas*, and we acknowledge that QS regulation can vary among different bacterial species under diverse environmental conditions. Our study focused on *B. glumae*, which exhibits unique QS regulation characteristics. This diversity among bacteria and their respective niches emphasizes the complexity of QS systems and the importance of studying them in various contexts. We have revised the manuscript to clarify the scope and context of our investigation, taking into account the variability in QS regulation among different bacteria.

Line 75: I do not understand what is meant by cooperativity here, especially in the example of *B. glumae*. The premise should be elaborated.

- In this context, cooperativity refers to the coordinated behaviors exhibited by bacteria to ensure the overall benefit of the population. Mutations in the *qsmR* gene disrupts the coordinated behaviors regulated by QsmR, potentially leading to a loss of these cooperative actions within the bacterial population.

Line 82: And most likely increases *tofl* expression.

- Reference 25 did not provide the quantitative analysis data of the expression of *tofl* in the *tofM* mutant.

Line 82-84: At high cell densities it just starts producing measurable quantities of AHL; this does not mean that the repression is 100% effective at LCD. One could then reason that the system has probably evolved to activate at very late growth stages.

- Producing measurable quantities of AHL at HCD does not imply 100% repression at LCD. Under laboratory conditions, the QS onset density for *B. glumae* was observed at $OD_{600} = 1.5$, equivalent to 1×10^9 CFU/mL, which is notably higher than in *V. fischeri* (ref. 24) or *P. aeruginosa* (ref. 13). QS-regulated gene expression was not detected until they reached the QS threshold (ref. 23). Based on these previous observations, it is evident that signal production and cell density are significantly correlated in *B. glumae*

Line 86 and afterward: I agree that the QS active metabolic phenotype will cause metabolic slowing, as the authors have shown this very nicely. However, saying it is important to avoid it at LCD implies that it does not cause metabolic slowing at HCD; so I find the point that the authors are trying to make here a bit confusing. The fact that QS does cause metabolic slowing is, however immensely interesting and does raise important questions in regards to the population growth dynamics.

- Previously we showed that QS acts as a metabolic brake to ensure homeostasis of metabolism when cells begin to mass (ref. 5). However, we did not intend to imply that metabolic slowing does not occur at HCD. Our intention in the highlighted sentence was to emphasize that the *tofM* mutation and early exposure of cells at low density result in slow growth, potentially due to QS-mediated metabolic slowing. We did not mean to suggest that the presence of stringent control of AHL production via *tofl* regulation by TofM serves solely to prevent metabolic slowing at LCD. The complexity of how QS modulates metabolism under different conditions is indeed intriguing and raises important questions about its impact on population growth dynamics.

Line 107: This indicated that *tofM* exerts a negative control on the average AHL production.

- We have revised the sentence in accordance with the reviewer's suggestion.

Line 150-153: You probably meant to say that you tested if the metabolic slowing of QS will select against QS in the fast-growing conditions. This is a very interesting result, that to me shows that in this case QS regulates a stress response phenotype and its implications are major; from the implications it has for "bet-hedging" to the fact that it shows QS regulation is irrelevant (and

therefore better off repressed) in fast-growing conditions. Please try to re-phrase, as your results are very important for the field of QS, since the way it is written I have trouble understanding what your actual interpretation of the results may be.

- We appreciate your insights. To provide additional clarification, it is crucial to understand that QS in *B. glumae* is not optional but essential for survival under specific laboratory conditions, such as growth in glucose-limited LB medium (ref. 23). In this environment, QS controls oxalate biosynthesis, which is vital for neutralizing the alkaline pH induced by deamination (ref. 23). Unlike the metabolic burden observed in the case of *P. aeruginosa* (ref. 28), the effects of the *tofM* mutant were not due to overproduction of public goods, but rather a result of the regulatory role of QS in metabolic homeostasis. It is important to highlight that the emergence of the *qsmR* mutant, which may be considered an optional “bet-hedging” strategy, occurs to relieve metabolic stress (ref. 26, 34). The spontaneous *qsmR* mutant indeed functions as a “cheater” in a broad sense, although it does not emerge under conditions necessitating the production of costly public goods (ref. 26, 34). As we have demonstrated in previous research, the spontaneous *qsmR* mutant arises as a response to alleviate metabolic stress (ref. 26, 34). Our investigation of the emergence of the *qsmR* mutant from the *tofM* mutant or WT with addition of AHL was performed to uncover the mechanisms underlying the TofM-mediated QS negative regulation system in *B. glumae*.

Line 201: You added exogenous signal to the culture, but did not actually study intra-species cross-talk. I agree, however, that exogenous signals in nature might in some cases originate from other species.

- While we acknowledge that exogenous signals may originate from other species in natural ecosystems, we did not examine the specific dynamics of interspecies crosstalk in this study.

Line 202: Are you implying that at high cell densities, no metabolic slowing is observed in your case?

- We do not intend to imply that metabolic slowing is absent at high cell densities but that most previous studies exploring QS-dependent phenotypes primarily focused on high-density cell cultures.

Line 202-207: It has been known, since the discovery of QS in *Vibrio fischerii*, that QS regulation is subject to additional regulation besides the concentration of QS signal. (Soto-Aceves et al., 2023). Please notice, that also in this article the authors refer to the concentration of AHL when talking about “the quorum threshold” and not to cell density, which just happens to correlate with it.

- We agree that the mere presence of a sufficient concentration of cognate AHL, as demonstrated in LuxI-LuxR-like circuits, does not act as a direct trigger for the expression of QS-dependent genes (Soto-Aceves et al., 2023). Our experimental results were consistent with this observation, as the *tofM* mutation or addition AHL alone was insufficient to activate QS-dependent genes in *B. glumae* (Figure S7). While the term “quorum threshold” is often associated with cell density, which naturally tends to be correlated with AHL concentration, our investigations revealed that cells at low density, when exposed to high levels of

externally supplied QS signals, experience metabolic imbalance and the emergence of noncooperative cells during growth. This highlights the critical role of negative regulation of AHL by TofM and the significance of this mechanism. In this context, we refer to the quorum threshold as the cell density, which is correlated with AHL concentration.

Line 209-211: The authors of ref. 23 do not show the mechanism of action of RsaM, only that it derepresses AHL production. The mechanism might actually be similar to what the authors have found for TofM in *B. glumae*.

- While there are indeed similarities in the amino acid sequence between RsaM in *Pseudomonas fuscovaginae* and TofM in *B. glumae*, it is worth noting that RsaM predominantly exerts its influence at the transcriptional level, while TofM acts primarily at the translational level.

Line 213-214; As mentioned before, in ref 23 the authors did not specify that RsaM inhibits transcription of *pfsI*, they only show that the transcription of *pfsI* is increased; the mechanism of repression might still be post-translational as in the case of TofM. Increased transcription might be due to the positive feedback loop of AHL QS systems.

- It should be noted that the mechanism of repression by RsaM in *P. fuscovaginae* may not be identical to that of TofM, as the transcription of *tofI* was not increased in the *tofM* mutant. In addition, for the positive feedback loop of AHL QS systems to take place, the presence of TofR, an AHL receptor, is required. However, it is worth mentioning that the level of TofR remains relatively consistent between WT and the *tofM* mutant.

Line 220: Unclear what the authors mean here

- Unlike RsmA of *P. aeruginosa*, which is known to be regulated by GacA/S and sRNA, TofM in *B. glumae* does not appear to be associated with any known two-component systems.

Line 226-228 At these densities measurable AHL quantities were produced and the system was activated. Please consider the positive feedback loop of AHL QS. It is most likely not cell density that directly triggered QS; as mentioned before.

- We acknowledge that cell density alone is not the sole trigger for QS. We reported previously that, in *B. glumae*, QS-regulated genes remained undetectable until the cell density reached at least 1×10^9 CFU/mL (ref. 23). Furthermore, we observed that the positive feedback loop of AHL-based QS may not develop until a minimum cell density of 1×10^9 CFU/mL is reached. This was supported by the lack of significant differences in the expression levels of *tofI* and *tofR* between WT and the *tofM* mutant under both LCD and HCD conditions.

Line 228-239: I fully agree with what I think the authors tried to convey here; that delaying virulence factor expression to late growth stages might significantly alter the pathogenicity dynamics of *B. glumae* and that this should be studied in a more pertinent "natural" environment; however I think that this paragraph should be re-written to better convey this intended meaning.

- We have incorporated the reviewer's feedback and restructured this paragraph to more effectively convey the intended meaning.

Line 240- 246: I don't understand what the authors meant to convey here.

- We intended to highlight the existing knowledge gap in the field of QS modulation. We proposed two possible explanations for this gap. First, there may be alternative, as yet undiscovered modes of action of QS modulation that differ from current models. Second, it is possible that mechanisms similar to the current models of QS modulation are widespread among QS bacteria, but they have not been studied or reported extensively. In essence, we intended to emphasize the complexity and diversity of bacterial communication systems and the need for further studies to explore and validate these possibilities.

Rest of the discussion: Please see the comment made for lines 150-153. Very often the authors raise rather general and non-specific questions and claim that these are understudied, however, it is very hard to evaluate this, because it is unclear to me what the specific question is in the first place. The authors show rather well that QS is active during late growth stages, and that QS causes metabolic slowing. The implications of these results are very interesting, however, the way the results are interpreted in the manuscript is confusing. What the authors clearly show in relation to the physiology of QS "active" cells is very important for the field of QS and merits clearer explanation.

- We have included the starting point of our study in the Introduction section, addressing density-related issues in *B. glumae*. We found that the threshold for QS onset in *B. glumae* significantly exceeded those in other well-studied QS bacteria. We have shown a correlation between cell density and AHL concentration in *B. glumae*, and have tried to elaborate the premise of our conclusions.

This study was performed to elucidate the mechanisms by which QS bacteria determine their quorum thresholds and to explore the effects on physiological and social attributes when faced with challenges in making such decisions. This study demonstrated the importance of control of *tofl* translation by TofM in defining a stringent quorum threshold for normal metabolism at LCD and maintaining cooperativity at HCD. We revealed the potential consequences of QS bacteria at LCD encountering an environment in which the QS threshold is exceeded by their natural neighbors. These findings raise very important questions about the physiological role of QS modulation at LCD, which has been overlooked in previous QS studies. We have added first two paragraphs in the Discussion section in the revised manuscript.

Comments on materials and methods:

Line 286: Specify what is Affymetrix?

- The company formerly known as Affymetrix has since become Thermo Fisher Scientific; it has been updated to Thermo Fisher Scientific in the revised manuscript.

Line 286: The antibiotics were added only when needed I presume? Same goes for C8 AHL probably?

- The antibiotics and C8-HSL were added only when needed.

Line 304-305 So the initial inoculum was 20x diluted culture of OD₆₀₀ 0.05? Is this true for all subsequent experiments?

- The 20x diluted culture with an initial OD₆₀₀ of 0.05 was specifically used for several experiments, including autoinducer assays, RNA isolation for qRT-PCR, Western blotting analysis, and real-time optical density measurements as shown in Fig. 3A. For the other experiments, we used an initial OD₆₀₀ of 0.05, as described in the Methods section.

Line 308: In the given reference there is no info on AHL extraction or C.violaceum assays.

- The reference in question was mistakenly cited as reference 31, but should have been cited as reference 32. This has been corrected and is now listed as reference 38 in the revised manuscript.

Line 363-364; You must have centrifuged the culture before filtering it?

- The culture was centrifuged before filtration, and have included this information in the revised manuscript.

Line 370: What was measured to produce the chromatography when separating the amino acids on the column? How was the concentration of amino acid determined?

- We measured the absorbance using both a fluorescence (FL) detector with an emission wavelength of 450 nm and excitation wavelength of 340 nm for OPA-derivatized samples, and an emission wavelength of 305 nm and excitation wavelength of 266 nm for FMOC-derivatized samples. In addition, a UV detector operating at 338 nm was used. To determine the concentration of amino acids, we used amino acid standards (agilent 5061-3330 and agilent 5062-2478). Comprehensive information regarding this methodology is presented in the manuscript.

372-281: Does this part just intend to say that you took 2 ml aliquots from growing cultures (and not from washed overnight "seed cultures), serially diluted them, and plated them? How does this assess viability of cells? Was the "seed" overnight culture washed also in the other experiments?

- In all our experiments, the overnight "seed" culture was consistently washed before preparing the subcultures. To assess the viability of cells and monitor the individual cell morphology within the growing cultures, we took 100- μ L aliquots from each sample without washing, performed serial dilutions, and spread the diluted samples on LB agar medium. This allowed us to observe the formation of single colony-forming units and assess colony morphology. We have revised the sentences accordingly for clarity.

Line 385 can you specify the page of the manual referenced as 31?

- We have provided the specific page of the manual now referenced as 39 in the revised manuscript.

Line 393-395 The cells were subcultured again after taking an aliquot? Probably you meant to describe the conditions as in lines 408-410?

- We intended to describe the conditions as outlined on lines 455–457. We have revised the sentences

accordingly in the revised manuscript.

Line 408-410 I assume that in every instance the seed culture and inoculation were prepared as is specified here. Please correct the other instances if this is so or maybe mention once how bacteria were cultured and that only an aliquot was taken for each analysis when needed.

- We have updated the sentence to state, "The seed culture and inoculation were prepared in the same manner when measuring catalase activity."

Line 419: By "honestly significant difference post hoc analysis" I presume the authors meant the Tukey "honest significant difference test"? Additionally, the p-value threshold of 0.05 does not mean the result was significant, it only indicates the false positive rate. The authors, however, might have chosen this threshold to be considered significant.

- The "honestly significant difference post hoc analysis" indeed refers to the Tukey honest significant difference test. We appreciate the clarification. We used a *P*-value threshold of 0.05, which signifies the false positive rate and is commonly employed to establish statistical significance, and we thank the reviewer for highlighting this point.

Overall comment:

The work done here is novel, impactful, and important for the field of QS; and I strongly recommend it for publication. However, the authors probably struggle with a language barrier, which I strongly feel should not invalidate their work. I also hope that the review did not come across as demeaning; this was not my intention. I am sure the authors understand their research very well, it's just that the way they expressed their results is confusing in places and I suggest to clarify.

References:

Platt TG, Fuqua C. What's in a name? The semantics of quorum sensing. Trends Microbiol. 2010 Sep;18(9):383-7. doi: 10.1016/j.tim.2010.05.003. Epub 2010 Jun 21. PMID: 20573513; PMCID: PMC2932771.

Spacapan, M., Bez, C., & Venturi, V. (2023). Quorum sensing going wild. iScience.

Hense, B., Kuttler, C., Müller, J. et al. Does efficiency sensing unify diffusion and quorum sensing?.

Nat Rev Microbiol 5, 230-239 (2007). <https://doi.org/10.1038/nrmicro1600>

Soto-Aceves, M. P., Diggle, S. P., & Greenberg, E. P. (2023). *Microbial Primer: LuxR-LuxI Quorum Sensing*. *Microbiology*, 169(9), 001343.

Smith, P., and Schuster, M. (2022). *Antiactivators prevent self-sensing in Pseudomonas aeruginosa quorum sensing*. *Proc. Natl. Acad. Sci. USA* 119, 2201242119.

Re: Spectrum03353-23R1 (Control of bacterial quorum threshold for metabolic homeostasis and cooperativity)

Dear Dr. Eunhye Goo:

In my opinion both Reviewers did an excellent job in providing precious suggestions to improve the quality of the manuscript, I am very grateful to them. I also appreciate your effort in editing the manuscript to address all the constructive criticisms raised by the Reviewers. Overall, I find the revised version of the manuscript suitable for publication in Microbiology Spectrum, so I am glad to communicate that your work has been accepted, and I am forwarding it to the ASM production staff for publication. Your paper will first be checked to make sure all elements meet the technical requirements. ASM staff will contact you if anything needs to be revised before copyediting and production can begin. Otherwise, you will be notified when your proofs are ready to be viewed.

Sincerely,
Giordano Rampioni
Editor
Microbiology Spectrum